# Master–Slave Control System for Virtual–Physical Interactions Using Hands

**DOI:** 10.3390/s23167107

**Published:** 2023-08-11

**Authors:** Siyuan Liu, Chao Sun

**Affiliations:** Test Automation and Control Institute, Harbin Institute of Technology, Harbin 150006, China; 1903010108@st.btbu.edu.cn

**Keywords:** master–slave control system, virtual–physical interaction, Kalman attitude fusion algorithm, stable virtual grasping algorithm

## Abstract

Among the existing technologies for hand protection, master–slave control technology has been extensively researched and applied within the field of safety engineering to mitigate the occurrence of safety incidents. However, it has been identified through research that traditional master–slave control technologies no longer meet current production and lifestyle needs, and they have even begun to pose new safety risks. To resolve the safety risks exposed by traditional master–slave control, this research fuses master–slave control technology for hands with virtual reality technology, and the design of a master–slave control system for hands based on virtual reality technology is investigated. This study aims to realize the design of a master–slave control system for virtual–physical interactions using hands that captures the position, orientation, and finger joint angles of the user’s hand in real time and synchronizes the motion of the slave interactive device with that of a virtual hand. With amplitude limiting, jitter elimination, and a complementary filtering algorithm, the original motion data collected by the designed glove are turned into a Kalman-filtering-algorithm-based driving database, which drives the synchronous interaction of the virtual hand and a mechanical hand. As for the experimental results, the output data for the roll, pitch, and yaw were in the stable ranges of −0.1° to 0.1°, −0.15° to 0.15°, and −0.15° to 0.15°, respectively, which met the accuracy requirements for the system’s operation under different conditions. More importantly, these data prove that, in terms of accuracy and denoising, the data-processing algorithm was relatively compatible with the hardware platform of the system. Based on the algorithm for the virtual–physical interaction model, the authors introduced the concept of an auxiliary hand into the research, put forward an algorithmic process and a judgement condition for the stable grasp of the virtual hand’s, and solved a model-penetrating problem while enhancing the immersive experience during virtual–physical interactions. In an interactive experiment, a dynamic accuracy test was run on the system. As shown by the experimental data and the interactive effect, the system was satisfactorily stable and interactive.

## 1. Introduction

Virtual reality technology, which is one of the most powerful human–computer interfaces, has seen rapid development in recent years. Its applications have been extensively employed in various fields, such as simulation training [1], medical rehabilitation [2], and manufacturing [3]. Hand-based virtual–physical interactions have long been a research focus, as humans heavily rely on their hands to interact with the external world [4]. Thus, using both hands for natural and flexible manipulation of virtual objects has become a primary choice for users. This mode of interaction is more intuitive and convenient than traditional methods, such as those involving a mouse, keyboard, or touch screen [5], and it aligns better with human behavioral habits and cognitive logic.

In recent years, research teams and organizations at home and abroad have explored a multitude of ways of capturing hand behavior and manipulating virtual objects. The main challenge is controlling and recreating a 3D hand model [6] of a user’s hand movements in real time with key data parameters, including the spatial position, orientation, and motion behavior. The current mainstream methods for motion data capture are primarily divided into two categories [7]: one uses an optical system, and the other involves a control system composed of various sensors. Although they are capable of independently capturing hand movements, optical systems fail to bypass certain limitations, such as spatial constraints, lighting conditions, and obstructions of motion markers [8]. These environmental factors limit data collection and the fluidity and integrity of motions [9]. Conversely, control systems based on multiple sensors cleverly circumvent these issues. Their compact design allows for attachment to the user, thus capturing motion data more accurately and reliably, and they are even applicable in complex working environments. To enhance the precision of data acquisition and the effectiveness of calibration techniques, data gloves have been proposed [10]; these primarily provide mapping between hand angles and the induction of a sensor, rather than using anatomy-based cross-effects [11]. In the design proposed by Saggio et al. [12], a novel sensor array structure was adopted to ensure alignment between the sensors at the fingertips, thus addressing the issues of friction and blockage. Spatial positioning and tracking schemes are vital innovations concerning VR, the interactions between computers and humans, and robot control [13]. By calculating the attitude data of an inertial measurement unit (IMU) installed on the torso, motion trajectories can be tracked in real time [14]. However, because the precision of the inertial components is not so high, data drift and cumulative errors occur over time. To overcome this phenomenon, a Kalman data fusion algorithm is commonly used to estimate the attitude angle.

Since the birth of robotic manipulators, master–slave control has been an important research topic, as it enables operators to remotely synchronize the control of a manipulator in various hazardous environments. The research on master–slave control started in the 1940s when researchers developed a robotic arm as a slave control device to replace the human arm. Currently, there are two mainstream approaches to master–slave control [15]: One utilizes optical systems, such as cameras and sensors, while the other employs a control system composed of multiple sensors, such as piezoelectric and inertial sensors [16,17].

In optical systems, the motion data of a hand are captured and tracked by using optical devices, such as cameras. Shamaie et al. proposed a generic hand-tracking algorithm that captures hand velocity through cameras and identifies dual-hand actions [18]. Manchanda et al. introduced a method for controlling mouse pointer movement by using simple gestures and a camera, and they incorporated a region algorithm for scaling the movement when the user moved away from the capture point [19]. Lai et al. presented a MEMS-based motion capture system design and demonstrated its ability to accurately reproduce human motion with an error of one degree [20]. Ceolini et al. proposed a sensor framework that integrated complementary systems and a fully neural morphological sensor fusion algorithm for gesture recognition, and they achieved significantly improved energy–latency products by 30 to 600 fold [21]. In optical systems, hand-tracking algorithms and optical devices are used to capture and recognize hand motions. Although optical systems can independently capture hand movements, there are still challenges that cannot be avoided, such as spatial limitations, lighting conditions, and occlusion of motion markers during capture, which restrict data collection, smoothness, and completeness and can prevent users from experiencing immersion. However, the advantage of optical systems is that they do not require additional devices to be worn and do not restrict freedom of movement, making them commonly used for indoor full-body motion capture.

With the advancement of technology and market demands, a wide range of sensors with complete varieties and reliable performance have flooded the market. Research teams and development institutions have successfully introduced various wearable devices as interfaces for human–computer interaction. Data gloves have been widely applied as hand interaction devices. A data glove [22] is a somatosensory human–computer interaction device that converts a user’s hand movements into electrical signals through motion data collection modules on the glove. The host computer consequently receives the transmitted signals, which are read and interpreted to control the robotic or virtual hand in order to replicate the hand movements of the user. The primary purpose is to establish a mapping relationship between the hand and the robotic fingertips and to overcome cross-effects based on anatomical constraints [23]. In [24], Silva et al. integrated bend sensors into a glove, developed a low-cost data glove based on the Arduino Uno microcontroller, and compared it with commercially available data gloves in the achievement of a virtual reality system. They designed and adopted a novel structure for a sensor array to ensure alignment between the top sensors of the fingers, thus resolving negligible friction and blocking issues and reducing deviation from the placement of the internal fingers. In [25], Youngsu et al. demonstrated the application of piezoelectric sensors as virtual reality interface devices by using flexible sensors made of PVDF material in a complete system. They compared the processed sensor outputs with real angles obtained from camera recordings and discussed the sensor performance. In [26], Mochamad et al. designed a virtual reality system for smartphones by using IMU (inertial measurement unit) sensors and flex bend sensors through the Unity engine. The test results showed a reduction in the rotational error to 1%, indicating that there was room for improvement. In [27], Saad et al. discussed an experimental study correlating the voltage output of a data glove with the bending angles of fingers. Through an analysis that involved polynomial regression, they allowed the angle-based interpretation of the voltage output, which is easier for users to understand. The experimental results demonstrated that the proposed method could accurately convert voltage into angles with 100% accuracy, thus providing a viable theoretical approach for future research.

This design is focused on intelligent hand wearable devices and, specifically, the design of hardware circuits and development of a software system. By fully utilizing resources, we derived a more reasonable hand posture solution model and applied suitable correction optimization methods. We designed a master–slave human–computer interaction system based on virtual reality involving hands, and we established an experimental platform based on Unity3D and Matlab in order to carry out a series of experiments and research. The second part of this article discusses the overall design scheme of the system and the design concept of the HDC data acquisition platform, and it introduces the working principles of the main functional modules. The third part discusses the analysis and simple processing of the raw data. The fourth part primarily discusses the deep processing of the data, including the filtering of the data and the posture solution, which make the data cleaner and easier to read. The fifth part discusses the system’s functionality and primarily introduces the design and control principles of a five-finger bionic mechanical hand, in addition to discussing the virtual grasping algorithm that we used in the virtual interaction process. The sixth part primarily introduces the debugging of the functions of the whole system, discusses the communication methods in each part of the system, and describes experiments on the system’s functionality. The seventh part concludes this study by summarizing the main content of this research and future research directions.

## 2. Structure of the System

The overall design of the master–slave virtual interaction system for hands mainly includes three parts: a hand data acquisition platform (HDT), a Unity3D-based virtual interaction platform, and a master–slave control platform. The hardware part of the master–slave virtual interaction system for hands is composed of a hand data acquisition platform and a master–slave control platform. The software part of the system mainly consists of a background data management platform and a virtual interaction system platform. The first step in master–slave virtual hand interaction is capturing the position, direction, and finger joint angles of a user’s hand in real time. The hand data acquisition platform uses a posture solution algorithm to analyze and calculate the motion information of the key parts of the hand in real time, transmit it to the background data management system for data fusion and filtering through wireless communication technology, accurately calculate the current spatial coordinates and motion direction, and transmit them to the Unity3D-based virtual interaction platform and master–slave control platform through serial port communication for synchronous mapping of the operator’s hand motion and posture. The Interactive process of the master–slave virtual hand interaction system is shown in Figure 1.

Hand Data Acquisition Platform (HDT)

After analyzing existing hand motion acquisition units, this study summarizes two mainstream technological routes. One is that of optical data acquisition systems, and the other is that of information perception control systems composed of various sensors.

In optical systems, hand motion data are collected and tracked by using optical devices, such as cameras. Hand motions are captured and recognized through hand-tracking algorithms and optical devices. A complete optical system consists of multiple sets of red optical lenses, reflective markers, POE switches, calibration frames, and cables, as shown in Figure 2. A typical capture system for optical motion employs several cameras with high speeds for tracking and monitoring specific attributes from various angles, which are then combined with algorithms that involve a skeleton solution to consummate the capture. In theory, the 3D location of a specific space can be determined at a particular moment if it is simultaneously observed by at least three cameras. When a camera photographs constantly with a significant frame rate, the trajectory may be derived from the sequence of images. Although an optical system can independently capture hand motions, some problems cannot be avoided, such as space limitations, lighting conditions, and the occlusion of motion markers during the capture process. These environmental impacts limit data collection and the smoothness and completeness of the motions, making it difficult for users to achieve an immersive experience. However, the advantage is that such systems do not require additional wearables, do not restrict human freedom of activity, and are commonly used for indoor full-body dynamic data capture.

A hand information perception and control system composed of sensors, which is also known as a “data glove”, is comprised of bend sensors, posture sensors, signal receivers, and microcontrollers, as shown in Figure 3. Its small size allows the system to be attached to the user and to adapt to complex environments, thus capturing motion data more flexibly and reliably. By utilizing the relationship between the output voltage and degree of bending of the bend sensors, the motion information of the fingers is calculated. Posture sensors are affixed to the main body parts, and posture signals are transmitted to a microcontroller through wireless transmission methods, such as Bluetooth, for posture calculation. The posture sensors obtain the posture information of the limbs through integrated inertial sensors, gravity sensors, accelerometers, magnetometers, and gyroscopes. In combination with the length information and hierarchical connections of the skeleton, the spatial positions of the joints are calculated. Because such systems are suitable for complex environments, they can capture motion data more accurately and reliably. The accuracy and stability of such systems do not decrease due to environmental factors.

In conclusion, a design scheme based on a hand information perception and control system composed of sensors as the data collection platform is adopted in our system to capture the angles of an arm’s posture in real time and to track hands’ motion trajectories in space.

2.Virtual and Real Interaction Platform Based on Unity3D

In virtual reality, interaction is indispensable. Interaction refers to when a program’s output generates results that are consistent with changes and effects that are expected according to a user’s input behavior. Unity3D is a comprehensive integration of professional virtual engines that allow users to easily create 3D games, real-time 3D animations, and other types of interactive content. Model movement can be controlled in Unity3D through certain script files, and some physical characteristics, such as collision, acceleration, and contact, can be simulated. In this study, we used Unity3D as our interactive program development platform to implement real-time mapping of hand movements. After an in-depth study of this platform, we found that its modeling capabilities are poor, and the software allows only the most basic geometric shapes. Complex models must be built in advance by using other software, so we chose to use the 3Dmax software for hand modeling and then imported the data into Unity3D for development.

3.Master–Slave Control Platform

The master–slave control platform is the link in the physical interaction with the entire system. The slave interaction device of this platform, that is, the robotic hand, should have the adaptive ability to grasp different objects and should be able to complete actions that are consistent with the movements of human hands. After studying the structure of robotic hands, we found that the key factors affecting the stability, convenience, and efficiency of grabbing are the number of fingers, the number of joints, and the number of degrees of freedom in the fingers. The advantages and disadvantages of robotic hands with different structures are shown in Table 1. The systems for the propulsion of industrial robots are classified as hydraulic, pneumatic, or electric based on the origin of their power. If necessary, they may be integrated to form a complex system. Table 2 outlines the characteristics of each fundamental system, as well as the relevant pros and cons.

After the above analysis, a structural design with five fingers, three joints, and five degrees of freedom was chosen for the slave interaction device in this project, and an electric drive system was adopted. To enhance the grasping effect of the robotic hand, a three-link coupled underactuated transmission structure was used. Each finger of this structure had a bionic structure with three phalanges and three joints. An SG90 servo was used at the finger–palm joint. This is a type of angle servo driver that is suitable for control systems in which the angle constantly changes and can be maintained. The other two joints were driven in a linkage-coupled manner. This not only made up for the shortcomings of the structure with five fingers, three joints, and five degrees of freedom, but also reduced the number of motors, thereby reducing the design’s complexity and enhancing the anti-interference ability of the robotic hand.

The pivotal first step in hand-based virtual–physical interaction is the real-time capture of the user’s hand position, hand orientation, and finger joint angles. Therefore, the authors constructed an HDT data glove and forearm posture detection module for the measurement of finger flexion, palm posture angles, and forearm posture angles. With these detailed data on hand movements, a virtual–physical interaction program that was developed in Unity3D was used to process the data packets. This established a corresponding mapping relationship between the user’s hand and the virtual hand’s fingers, palm, and arm, thus defining the motion trajectory of the virtual hand. Consequently, this enabled the virtual hand to execute interactive movements in harmony with the user’s hand. The overall structure of the system is illustrated in Figure 4.

Data acquisition is crucial to this system. The constructed HDT data glove employed flexible sensors [28] and posture sensors as hand data collection units, thus facilitating the mapping of virtual palm movements. In the forearm posture tracking system, posture sensors were also used in the forearm posture detection module. This module’s primary function was to capture real-time forearm posture angles and track the hand’s motion trajectory in space. The data captured by the sensors were processed by the glove’s control unit and the forearm posture module through a Kalman posture fusion algorithm and by using data-amplitude-limit jitter reduction. The acquired hand motion data were then transmitted to the host computer via Bluetooth and Wi-Fi (wireless communication) for data fusion analysis, and the current spatial coordinates and motion direction were then calculated.

Figure 5 displays the hardware framework of the HDT, which was composed of five parts: the main control module (atmega328p microcontroller), the power module (AMS1117-5-3.3 voltage regulation module), the data collection module (five-way-flexible sensors, attitude sensor), the monitoring module (current drive chip TB6612), and the communication module (Bluetooth HC-05).

The primary workflow of the framework is shown in Figure 5.

(1) An external 3–5 V power supply was connected to an AMS1117-5-3.3 voltage regulation circuit module. The principle of the AMS1117-5-3.3 voltage regulation circuit is shown in Figure 6. The voltage regulation circuit module transformed the power supply into a stable internal power supply with an output of 5 V (or 3.3 V).

(2) The LM1117-5 voltage regulation circuit module provided a stable power supply for the mpu9250 attitude sensor, the flexible sensor, and the main control system, while the main control system monitored whether the current was stable by using the pulse wave output by the TB6612 current drive chip. The principle of the circuit of the current drive system is shown in Figure 7.

(3) The attitude sensor and the flexible sensor separately transmitted the data that were collected to the main control module. The principle of the main control module is shown in Figure 8. The attitude sensor input data into the microcontroller through the IIC interface, while the flexible sensor output voltage signals through the connection to the IO port of the microcontroller.

(4) The main control system (atmega328p microcontroller) integrated and filtered the data that were collected from the attitude sensor and the flexible sensor and transmitted them to a PC for processing through the HC-05 Bluetooth module.

To facilitate subsequent testing of the practicality and reliability of this hardware system, we simply built a 1.0 version of the HDT based on this description of the hardware framework, as shown in Figure 9.

In this framework, the collection of hand motion data was crucial. We used five-way-flexible sensors and an attitude sensor as our data collection module. The five-way-flexible sensors were attached to parts of the fingers to detect the degrees of finger bending. When the fingers were bent, the controller’s microcontroller was connected to the sensor ends through the I/O port. When the voltage changed, the microcontroller was able to read the voltage value through the I/O port, and the degree of finger bending was calculated and quantified in analog quantities. The attitude sensor was placed in the center of the back of the hand to detect real-time hand posture angle data. The data were input into the IIC pin of the microcontroller through the IIC interface of the sensor, thus enabling the controller to obtain data on the hand posture angles. The MPU9250 is an integrated accelerometer, magnetometer, and gyroscope sensor; therefore, it reduced the size and power consumption of the system and allowed the inter-axis difference caused by the presence of multiple sensors in the system to be effectively avoided.

In the hand posture tracking system, we also developed two upper- and lower-arm modules based on the MPU9250 attitude sensor. They were placed in the central position of the upper and lower arms. Their serial port interfaces were connected with the serial port interface of the WIFI chip, thus allowing the MPU9250 data packet to be transformed into a wireless signal through the WIFI chip. The main function of this module was capturing the posture angle of the upper and lower arms in real time and tracking the motion trajectory of the hand in space.

The posture sensor used in this system was the MPU9250, which consisted of two parts. Three-axis accelerometers and three-axis gyroscopes made up one component. The second component was the AK8963 from AKM. Therefore, the MPU9250 enabled the tracking of motion with nine axes. Its unique motion digital processing engine (DMP) was located inside the MPU9250 and could directly process data, thus reducing the number of tasks for the main control chip and requiring only the transmission of the three-axis gyroscope, accelerometer, and magnetometer values to the DMP. The IIC approach was able to directly provide all nine axes’ data. Integrated design, movement integration, and clock calibration eradicated the time-consuming choice of complicated processors and peripheral expenses, thus guaranteeing the best performance. Additionally, this device offered an IIC interface to allow compatibility with comparable sensors, such as connecting pressure sensors.

The posture angle solution refers to the angle acquired by accumulating and integrating the acceleration, angular velocity, and magnetic field intensity (all with three axes) via the attitude sensor. The results are the roll angle (roll), pitch angle (pitch), and yaw angle (yaw). The coordinate system used for the posture angle solution was the east–north–sky coordinate system. The module was placed in the positive direction, as shown in Figure 10, with the left as the x-axis, the front as the y-axis, and the top as the z-axis. The order of rotation of the coordinate system when the Euler angle represented the posture was defined as z-y-x, i.e., first rotating around the z-axis, then the y-axis, and, finally, the x-axis.

## 3. Data Analysis

To obtain the motion data of the fingers, palms, and arms, a single-chip microcomputer was employed to analyze and calculate the data acquired by the flexible sensors and posture sensors.

### 3.1. Analysis of the Bending Degrees of the Fingers

A five-part flexible sensor whose output voltage varied with the bending deflection and was positively correlated with finger flexion was attached to the fingers. The correlation between the output voltage and finger flexion is illustrated in Figure 11.

The output voltage of the flexible sensors ranged from 1000 mv to 4000 mv. Thus, it was assumed that the data read by the single-chip microcomputer were from *AF*_1_ to *AF*_5_.

Though a byte is in the numerical range of 0–255, the most suitable range for mapping is from 50 to 200; then, a byte can be used to represent a finger’s flexion.
$$a_{n}=AF_{n}/20$$
$$a_{n}\in (50,200)$$

*AF*_1.1_, the data of the first joint in the thumb, ranges from −100° to 50° in Unity. Thus, Equation (1) (*AF*_1.1_ = *a*_1_ − 150) can be used to map this range.
*Ltz* = (*lfresult1[1]* − 150) + *UI.lthum_a*(1)

Here, *ltz* refers to *AF*_1.1_, *lfresult1[1]* refers to *a*_1_, and *UI.lthum_a* refers to the deviation value of the slider, which was manually set in the interface of Unity so as to adjust the finger model. Naturally, the data of the remaining 13 joints can be calculated in the same way. The only difference among these data lies in the range of flexion of the joints.

### 3.2. Analysis of the Eulerian Angles of the System

Posture sensors were attached to the back of the hand and the middle of the upper and lower arms so as to acquire real-time posture angles of the palm and the upper and lower arms.

The Eulerian angles refer to the roll, pitch, and yaw [29], whose calculation was based on the data acquired by the posture sensors (MPU9250) and on the acceleration, angular velocity, and magnetic field intensity on the three axes [30].

With the accelerometer, the initial roll *β* and pitch *θ* can be calculated on the basis of Equations (2) and (3).
(2)$$\beta =\arcsin\left(-\frac{a_{x}}{g}\right)$$
(3)$$\theta =\arctan\left(\frac{a_{y}}{a_{z}}\right)$$

$a_{x}$, $a_{y}$, and $a_{z}$ refer to the acceleration values on the three axes. Since the accelerometer could not sense the rotation angle on the z-axis, the magnetometer, which was used to measure the magnetic induction intensity, needed to be employed to help calculate the yaw.

(1) When the magnetometer was parallel to the ground, the yaw could be calculated on the basis of Equation (4).
(4)$$\psi =\arctan\left(\frac{m_{y}^{n}}{m_{x}^{n}}\right)$$

$m_{y}^{n}$ and $m_{x}^{n}$ refer to the magnetic field intensity on the *x*-axis and *y*-axis.

(2) When the magnetometer was not parallel to the ground, tilt compensation could be adopted to reduce errors and help calculate the yaw. Due to the angle between *n* (geographic) and *b* (carrier) that was included when using these as the coordinate systems, Equation (5) was necessary for the geographic coordinate system’s magnetic field.
(5)$$\left[\begin{array}{l} m_{x}^{n}\\ m_{y}^{n}\\ m_{z}^{n} \end{array}\right]=C_{b}^{n}\left[\begin{array}{c} m_{x}^{b}\\ m_{y}^{b}\\ m_{z}^{b} \end{array}\right]$$

$m_{x}^{b}$, $m_{y}^{b}$, and $m_{z}^{b}$ refer to the magnetic field intensity measured on the *z*-axis. $C_{b}^{n}$ refers to the cosine orientation matrix.
(6)$$C_{b}^{n}=\left[\begin{array}{ccc} \cos\beta & \sin\beta \sin\theta & \sin\beta \cos\theta \\ 0 & \cos\theta & -\sin\theta \\ -\sin\beta & \cos\beta \sin\theta & \cos\beta \cos\theta \end{array}\right]$$

With Equations (5) and (6), the tilt compensation formula can be deduced.
(7)$$m_{x}^{n}=m_{x}^{b}\cos\beta +m_{y}^{b}\sin\beta \sin\theta +m_{z}^{b}\sin\beta \cos\theta$$
(8)$$m_{y}^{n}=m_{y}^{b}\cos\theta -m_{z}^{b}\sin\theta$$

With $m_{y}^{n}$ and $m_{x}^{n}$, the compensated magnetic field intensity and yaw can be calculated on the basis of Equation (4).

## 4. Data Calibration

### 4.1. Calibration of the Bending Degrees of the Fingers

The data output by the flexible sensor were the values of the voltage, which could experience small jumps. Therefore, a cumulative jitter elimination filtering algorithm was used. When the collected data continuously jumped beyond the set valve, this indicated that the overall numerical range of the element had changed. Therefore, a new element was assigned to the output value. If the change did not exceed the threshold or did not continuously exceed the threshold, this indicated that the change in the data was in a small range of noise, which was filtered out, and the output remained at the original value. This algorithm can effectively remove small-range jitters in data, thus making the data smoother. For possible high-amplitude errors, a limited-amplitude filtering algorithm was adopted to identify and eliminate noise that exceeded a certain amplitude and to take the true value of the data within the threshold. The original data were filtered through these two filtering algorithms to make them more realistic and stable. A comparison of the effects before and after filtering is illustrated in Figure 12.

### 4.2. Calibration of the Eulerian Angles of the System

If the angle data would directly calculated from the acceleration data by using Kinematics, there would be a problem due to poor dynamic accuracy because the dynamic data of an accelerometer are not accurate when an object moves, but the advantage is that there is no long-term drift or error accumulation [31]. Dynamically and precisely measuring a three-axis attitude angle is possible by integrating the angular velocity obtained with a gyroscope, but the disadvantage is that error accumulation and drift occur, and the accumulated error increases with time. The Kalman filter is a data fusion algorithm that combines the data of an accelerometer, magnetometer, and gyroscope to calculate the attitude, and it combines the advantages of all of these sensors to accurately measure the attitude in a dynamic environment. The algorithm’s process is illustrated in Figure 13.

$x_{k}$ refers to the state vector at moment *k*. $y_{k}$ refers to the observation vector at moment *k*. ***A*** refers to the state-transition matrix from moment *k* to moment *k −* 1. **H** refers to the gain matrix from the state vector to the observation vector. *q* and *r* refer to the input noise and the observation noise, whose covariance matrices are represented by ***Q*** and R. Supposing that ${\hat{x}}_{k}^{-}$ is the estimated value at moment *k* and ${\hat{x}}_{k}$ is the adjusted value at moment *k*, then the following is true.

Assuming the estimated value at time *k* and the corrected value at time *k*, then
(9)$${\hat{x}}_{k}^{-}=A{\hat{x}}_{k-1}+q$$
(10)$${\hat{x}}_{k}={\hat{x}}_{k}^{-}+K\left(y_{k}-H{\hat{x}}_{k}^{-}\right)$$
where *K* refers to the Kalman gain matrix, which is a key factor in accurately estimating the state.
(11)$$K_{k}=P_{k|k-1}H^{T}{\left[HP_{k|k-1}H^{T}+R\right]}^{-1}$$

Here, $P_{k|k-1}$ refers to the pre-estimated error covariance matrix.
(12)$$P_{k|k-1}=AP_{k-1}A^{T}+Q$$

$P_{k}$ refers to the error covariance matrix at moment *k*.
(13)$$P_{k}=\left(I-K_{K}H\right)P_{k|k-1}$$

I refers to the unit matrix.

In the source code program, the initial values of the coefficients of the Kalman pose fusion algorithm were as follows:$$A=\left[\begin{array}{cc} 1 & -0.01\\ 0 & 1 \end{array}\right],\;H=\left[\begin{array}{ll} 1 & 0 \end{array}\right],\;{\hat{X}}_{0}=\left[\begin{array}{l} 0\\ 0 \end{array}\right],\;P_{0}=\left[\begin{array}{ll} 1 & 0\\ 0 & 1 \end{array}\right],\;Q=\left[\begin{array}{cc} 0.2 & 0\\ 0 & 0.1 \end{array}\right],\;R=2$$

The attitude estimation model derived above was implemented by modifying certain parameter codes—specifically, matrices **A** and **H**—as well as the dimensions of other matrices. The results of the implementation are shown in Figure 14, where the blue line on the left represents the observed waveform of sinusoidal data with an increasing amplitude and overlaid with random noise. The red line represents the waveform of the data after they were filtered with the Kalman filter. The right side of the figure shows a localized comparison that clearly indicates that the filtered data were smoother than the original waveform.

Here, c***Q*** and c***R*** correspond to ***Q*** and ***R***, respectively, in the equation. By reducing the ratio of c***Q***/c***R***, the waveform was obtained, as shown in Figure 15, which demonstrates both conciseness and professionalism.

In Figure 16, the filtering results closely match the observed values, indicating that the Kalman filter allowed high confidence in the observations at this point.

The choice of observations is crucial in the Kalman filtering algorithm. The size of the observation error directly affects the effectiveness of Kalman filtering. To enhance the system’s resistance to interference, ensure the control precision, enhance the stability of state observations, and mitigate the effects of external forces on observations, the Kalman fusion complementary filtering algorithm was used in the attitude solution algorithm of this system. We chose the attitude angle from the fusion of the complementary accelerometer and magnetometer filters as the state observation variables for the Kalman algorithm, and we utilized the gyroscope’s data and noise to establish the state prediction equation for iteration. This allowed for the real-time acquisition of highly accurate and trustworthy attitude angles. The structure after the fusion of the two algorithms is shown in Figure 17.

The fundamental structure of the complementary filtering algorithm is depicted in Figure 18, and the relevant formula is presented in Equation (14).
(14)$$\left\{ \begin{array}{l} \phi =\phi_{g}+k(\phi_{am}-\phi_{g})\\ \theta =\theta_{g}+k(\theta_{am}-\theta_{g})\\ \psi =\psi_{g}+k(\psi_{am}-\psi_{g}) \end{array}\right.$$

Here, $\phi_{g}\;\theta_{g}\;\psi_{g}$ and $\phi_{am}\;\theta_{am}\;\psi_{am}$ represent the attitude angles that were calculated from the gyroscope, accelerometer, and magnetometer, respectively, while $\phi_{}\;\theta_{}\;\psi_{}$ signifies the attitude angle after fusion.

To further verify the stability and accuracy of the posture data, we fixed the attitude sensor on a DC motor (MAXON RE35) with an encoder, rotated it 45° around its x-, y-, and z-axes, respectively, and obtained a set of sensor output data, as shown in Table 3.

The dynamic output waveform of the 45° rotation around the x-axis in the positive direction and the return to the initial state is shown in Figure 19. The motor was set to start rotating 45° in the positive direction around the x-axis (roll angle) at moments 0–3, to return to the initial state at moments 3–6, and to stay at moments 6–8. From the waveform, it can be seen that the process of change in the angle was stable and smooth, with no distortion; moreover, it remained relatively stationary in the direction of the y-axis (pitch angle) and z-axis (yaw angle), and the output waveform was stable within ±0.2°. In the tests with two posture sensors, the outputs of the roll, pitch, and yaw were stable at ±0.1°, ±0.15°, and ±0.15°, respectively, signifying the suitable IMU sensor compatibility of the algorithm with respect to the reduction in noise and precision [32]. These data can be used to map hand posture actions in three-dimensional space.

Euler angles are the most familiar form of angles to many people. Their three axes are coupled and display independent variations only in small angles. For larger values, the attitude angles change in a coupled manner. For example, when the x-axis approached 90 degrees, even if the attitude rotated solely around the x-axis, the y-axis angle also underwent significant changes. This is an inherent characteristic of the representation of Euler angles, and it is known as gimbal lock. In Unity3D, Euler angles also suffer from gimbal lock issues, leading to spasms and halts when two axes are on the same plane. To avoid gimbal lock issues, angles must be represented in quaternion form, as shown in Equation (15):(15)$$q=\lambda +P_{1}i+P_{2}j+P_{3}k$$
where λ represents the scalar part, while $P_{1}i+P_{2}j+P_{3}k$ stands for the vector part. The conversion from Euler angles into quaternions is shown in Equation (16).
(16)$$q=\left[\begin{array}{l} \lambda \\ P_{1}\\ P_{2}\\ P_{3} \end{array}\right]=\left[\begin{array}{l} \cos(\phi /2)\cos(\theta /2)\cos(\psi /2)+\sin(\phi /2)\sin(\theta /2)\sin(\psi /2)\\ \cos(\phi /2)\sin(\theta /2)\cos(\psi /2)+\sin(\phi /2)\cos(\theta /2)\sin(\psi /2)\\ \cos(\phi /2)\cos(\theta /2)\sin(\psi /2)-\sin(\phi /2)\sin(\theta /2)\cos(\psi /2)\\ \sin(\phi /2)\cos(\theta /2)\cos(\psi /2)-\cos(\phi /2)\sin(\theta /2)\sin(\psi /2) \end{array}\right]$$

In this equation, ϕ,θ,ψ denote the rotation angles around the x-, y-, and z-axes, respectively. They are expressed by using Tait–Bryan angles—specifically, the roll, pitch, and yaw.

## 5. Design of the Mechanical Hand and Realization of Virtual–Physical Interaction

### 5.1. Design of the Mechanical Hand

To meet the requirements of the system’s functions and the virtual hand’s compatibility, a three-link coupled structure was employed to design the five-finger mechanical hand that was used with the system. Its overall structure is illustrated in Figure 20. Its drive mode was relatively rigid, and its fingers had great force for grabbing objects, making it highly self-adaptable to the target object. The index finger’s process of adaptively grabbing objects is illustrated in Figure 21.

To better control the mechanical hand, a mathematical model was established. By doing so, motion planning was conducted, and the restraint of motion on the end effector of the mechanical hand when contacting the target object was discovered. To analyze this with Kinematics, the virtual hand was simplified into a link structure, and each phalanx of the fingers was a link connected by a joint, as shown in Figure 22.

The characteristics of the link structure in Kinematics can be interpreted with D–H theory. That is to say, any position of the hand in the local coordinates can be interpreted as the following matrix A in global coordinates.
$$\begin{array}{cc} A & =\left[\begin{array}{cccc} \cos\theta_{i} & -\sin\theta_{i}\cos\alpha_{i} & \sin\theta_{i}\sin\alpha_{i} & \alpha_{i}\cos\theta_{i}-d_{i}\sin\theta_{i}\\ \sin\theta_{i} & \cos\theta_{i}\cos\alpha_{i} & -\cos\theta_{i}\sin\alpha_{i} & \alpha_{i}\sin\theta_{i}+d_{i}\cos\theta_{i}\\ 0 & \sin\alpha_{i} & \cos\alpha_{i} & 0\\ 0 & 0 & 0 & 1 \end{array}\right]\\ \; & =\left[\begin{array}{cccc} \cos\theta_{i} & -\sin\theta_{i} & 0 & L_{i}\\ \sin\theta_{i} & \cos\theta_{i} & 0 & 0\\ 0 & 0 & 1 & 0\\ 0 & 0 & 0 & 1 \end{array}\right] \end{array}$$

The parameters of D–H are presented in Table 4. 

We built a simulation model in SolidWorks by using ABS plastic material for 3D printing, as this is the most commonly used printing material in consumer-grade 3D printing. For the control board of the robotic arm, we chose the XO control board to control the robotic arm, and the communication method was that of serial communication. The five-finger bionic robotic hand after all parts were assembled, wired, and powered on is shown in Figure 23.

Currently, two prevalent mapping methods are in use: joint angle mapping and fingertip position mapping. Fingertip mapping projects the operator’s fingertip position onto a robotic hand to enable accurate mirroring of the operator’s hand movements in three-dimensional space. To ensure smooth mapping, it is necessary to establish corresponding palm and joint coordinate systems between the human hand and the robotic hand, thus ensuring one-to-one correspondence. Compared with that in fingertip mapping, the computation process in joint angle mapping is more streamlined, as only proportional mapping is required to generate movement instructions for the robotic hand. The effects of both control methods can meet the requirements of work environments that do not require precise control. In this system, we selected joint angle mapping as our motion mapping method. The implementation process for master–slave control is illustrated in Figure 24. Before executing master–slave control, the operator first needs to wear a data glove and perform maximum-extension and fist-clenching movements. The purpose of this is for the data management software to record the maximum (*A_Umax*) and minimum (*A_Umin*) bending angles of the five finger joints of the operator. Given the structural constraints of our robotic hand, the range of motion of each finger joint was predetermined; thus, we also set the maximum (*A_Cmax*) and minimum (*A_Cmin*) bending angles of the robotic finger joints. At a certain moment, if the bending angles of the operator’s hand and the robotic hand were *A_Umid* and *A_Cmid,* respectively, then *A_Cmid* could be obtained by using Equation (17).
(17)$$A\_C_{mid}=\frac{A\_U_{mid}-A\_U_{\min}}{A\_U_{\max}-A\_U_{\min}}\times (A\_C_{\max}-A\_C_{\min})+A\_C_{\min}$$

### 5.2. Realization of Virtual–Physical Interaction

#### 5.2.1. Realization of Virtual Grasping

The tracking device and the driven virtual hand were unidirectionally coupled, which meant that the former could drive the latter, but the latter could not drive the former. As a result, there was no control with a closed loop between the user’s hands and virtual hand. As a result, the latter’s fingers could penetrate into a virtual object, and the engine was unable to address the issue, as tracking data powered the hand without taking any simulation instabilities into account. To address the issue of penetration, we devised a proxy model that was linked to the virtual hand and was capable of providing a visual reaction to its environment. Collision and grabbing occurred between the 3D virtual object and the proxy hand. The process is shown in Figure 25.

This problem is seen as the problem of penetration depth [33]. Figure 26 shows a penetration analysis of the proxy hand from both the geometric and graphical perspectives. The penetration depth is a measure of the degree of mutual penetration between collision penetration models. It is widely used in path planning, physics-based simulation, haptic simulation, and computer-aided modeling. The classifications of penetration depth are translational (PDt) and generalized (PDg). The primary distinction is that the latter can detach overlapping interactive objects at the same time. We assume that there are two models that overlap, A and B, with A moving and B remaining stationary. Object A’s initial local coordinates are identical to those of the global system O. Objects A and B can be separated by using a fixed motion. To calculate the minimum penetration depth for objects A and B, PDg is in the six-DOF space. The depth measurement is provided in Equation (18) [34]:(18)$$PD_{g}^{\sigma }(A,B)=\left\{\min\{\sigma_{A}(q,0)\}||\mathrm{interior}(A(q))\cap Β=\emptyset ,q\in F\right\}$$
where **q** is the initial collision state of **A**, **F** is the non-collision space, and σ is the measure of the distance between the two poses of **A** in a six-degree-of-freedom space. Here, we refer to it as the target norm.

Usually, any distance metric can be chosen to define σ. Here, the target norm is chosen as the unit of distance measurement, as it has a compact analytical expression and is invariant to the reference coordinate system. For object A, the target norm in positions $q_{0}$ and $q_{1}$ and in position $q_{1}$ is defined as
(19)$$\begin{array}{cc} \sigma_{A}(q_{0},q_{1}) & =\frac{1}{V}\int_{x\in A}{\left(x(q_{0})-x(q_{1})\right)}^{2}\\ \; & =\frac{4}{V}(I_{xx}q_{1}^{2}+I_{yy}q_{2}^{2}+I_{zz}q_{3}^{2})+q_{4}^{2}+q_{5}^{2}+q_{6}^{2} \end{array}$$
where x(q) refers to the point where A is in position **q**. $\left[q_{1},q_{2},q_{3}\right]$ refers to the quaternion vector—or the quaternion vector of the rotation transformation—which marks the relative azimuth difference between $q_{0}$ and $q_{1}$. $\left[q_{4},q_{5},q_{6}\right]$ refers to the relative position difference between $q_{0}$ and $q_{1}$. **V** refers to the volume of A, and I refers to the diagonal element of the inertial tensor matrix of A.

The virtual model was mapped to an auxiliary model for a dynamic simulation of the two objects. The tracking of the initial motion of the virtual hand and the auxiliary hand could be controlled on the basis of the acquired data and the contact state. In the interaction, the auxiliary hand—the main interactive hand—was used to test whether there was an effective contact. The whole interaction can be interpreted in terms of four states: the contactless state, contact state, penetration state, and release state (Figure 27).

(1)Contactless State (S1): The hand is not in contact with the manipulated object.(2)Contact State (S2): Starting to separate from the virtual hand, the auxiliary hand remains at the contact spot.(3)Penetration State (S3): The auxiliary hand grasps the object that is being manipulated as the virtual one penetrates it.(4)Release State (S4): The auxiliary hand moves away with the virtual one, thereby releasing the object that is being manipulated.

If the hand and the manipulated object are not in contact, the auxiliary hand moves with the virtual hand. However, once the two are in contact, the auxiliary hand stops moving with the virtual hand and remains relatively stationary in the contact spot, even when the object is penetrated by the virtual hand. When releasing, the auxiliary hand continues to move with the virtual hand. With the release state comes the contactless state, and thus, the grabbing interaction circulates. The difference in the interactions of the virtual hand and the auxiliary hand is illustrated in Figure 28.

#### 5.2.2. Realization of Stable Grasping

The method of relative thresholds was employed to determine whether an object could be grasped, thereby maximizing the realism of actual life. First, we located each object’s bounding box that was aligned with the axis; consequently, we determined the face of the bounding box that was most proximate to the hand. By calculating the diagonal length of this face, we could determine a real-time relative threshold in order to map the size of the graspable face of the hand onto a threshold range from 0 to 1. This is illustrated in Figure 29.

We assume that x represents the size of the face. Then, *x* can be mapped from the object size range $\left[O_{\min},O_{\max}\right]$ to the threshold range $\left[T_{\min},T_{\max}\right]$. Therefore, the first step is computing the location ρ of x within the object size range $\left[O_{\min},O_{\max}\right]$, as shown in Equation (20):(20)$$\rho =\frac{x-O_{\min}}{O_{\max}-O_{\min}}$$
where *x* is the diagonal length of the *face*, $O_{\min}$ is the diagonal length of the smallest face of the object, and $O_{\max}$ is the diagonal length of the largest face of the object.

Having obtained ρ, we should then compute the value of *x* within the threshold range $\left[T_{\min},T_{\max}\right]$, which represents the threshold required for the object, as demonstrated below:(21)$$\lambda =\rho (T_{\max}-T_{\min})+T_{\min}$$
where λ is the threshold required to grasp the object, $T_{\max}$ is the maximum threshold, and $T_{\min}$ is the minimum threshold. An object is only grasped when the grasping posture satisfies the threshold relative to the size of the object, thus ensuring that users can grasp more naturally and stably.

The stable grasping of an object can also be determined based on a physics-based method, i.e., by using the calculation of the force during the interaction process as a factor in assessing the stability of the grasp. From a mechanical and dynamic perspective, the balance between the force and the torque acting on the manipulated object is one of the conditions for stable grasping. Suppose that the virtual object and the virtual hand have *n* contact points, the normal force at the *i*-th contact point is set to F(i), the frictional force is f(i), and the size of the gravitational force of the object is $F_{g}$. Therefore, the condition for stable grasping can be expressed by Formula (22):(22)$$\begin{array}{l} \left|\sum F\right|=\left|\sum_{i=1}^{n}F(i)+\sum_{i=1}^{n}f(i)+F_{g}\right|=0\\ \left|\sum M\right|=\left|\sum_{i=1}^{n}F(i)\times r_{i}+\sum_{i=1}^{n}f(i)\times r_{i}\right|=0 \end{array}$$
where ∑M,∑F represent the combined torque and combined force, and $r_{i}$ represents the radial vector between the contact point and the center of mass of the object.

When the above equilibrium conditions are satisfied, stable grasping can be achieved. Subsequently, the coordinate system of the virtual object merges with the root coordinate system of the virtual palm, thus enabling the virtual hand to translate or rotate the object.

## 6. System Debugging and Experimentation

### 6.1. Establishment of the Data Platform

A back-end data management platform served as a data “relay station”, with its primary functions encompassing data reception, monitoring, storage, processing, fusion, and transmission. This platform was built on the MFC framework and was designed by using a multithreaded approach, which ensured the parallel operation of its functions. The purpose of developing this platform was to provide operators with a more intuitive and clear way to observe and manage data, merge and package the data, and then send them to the master–slave control platform and the virtual interaction platform. The data management software is illustrated in Figure 30.

It is well known that the size and shape of each person’s hand vary. To allow different users to have a good interactive experience, this data management platform had data correction functions, such as “initial attitude calibration” and “initial finger calibration”. Their purpose was to record the range of motion of different users’ fingers and palms and normalize them, thus enabling the hand motion data of each user to fit the system’s data format and solving the data matching problem for different users. The specific calibration methods were as follows.

(1) Initial finger calibration: The user wore the data glove and performed fist-clenching and finger-stretching movements five times. The system recorded the range of finger movement and normalized it, as shown in Figure 31.

(2) Initial palm attitude calibration: While wearing the data glove, the user followed the operation shown in Figure 32. Clicking on “end attitude calibration” completed the hand attitude calibration.

It is worth noting that data transmission was a key function of this platform. To enable the cross-platform transfer of motion data from the data acquisition platform to the virtual–physical interaction platform, we utilized the User Datagram Protocol (UDP). UDP sockets facilitate data transfer between programs and support data transmission between two distinct platforms. For this instance, virtual port numbers were used to send data to memory addresses, and Unity constructed a listener for data retrieval. Upon the establishment of a connection, real hand movements could be captured and analyzed, thereby enabling the mapping of the operator’s hand movements in a virtual environment by using a virtual hand. The overall data packet of the software was as follows.

0x55 + 0x53 + RollL + RollH + PitchL + PitchH + YawL + YawH + 0x55 + 0x53 + RollL + RollH + PitchL + PitchH + YawL + YawH + 0x55 + 0x53 + RollL + RollH + PitchL + PitchH + YawL + YawH + 0xaa + a1 + a2 + a3 + a4 + a5 (30 bit). The specific definitions are as follows.

This software packaged data into 30 bit data packets, which included posture angle and finger-bending data. The first 24 bits consisted of data from three posture sensors, with each contributing 8 bits of data. The data frame began with the header “0x55, 0x53”, immediately followed by six bits of data that represented the roll, pitch, and yaw. Here, “*RollL*, *RollH*, *PitchL*, *PitchH*, *YawL*, *YawH*” were the raw data from the sensor’s IIC interface and not the direct angle values. The last six bits represented finger curvature data, starting with the data frame header 0xaa and ending with the curvature of the five fingers.

### 6.2. Testing the Interactive Virtual Reality System 

The commonly encountered “OK” and “V-sign” gestures were used as our experimental targets. The interaction is shown in Figure 33.

As can be seen in Figure 34 and Table 5, the virtual hand was able to accurately and authentically interact with the human hand; the process was smooth and continuous, and the movements remained consistent. Figure 5 and Figure 6 validate this process; we observed that the “OK” gesture and the “V-sign” gesture were completed within three seconds. Here, a_1_–a_5_ correspond, respectively, to the middle finger, little finger, ring finger, thumb, and index finger. During the movement process, there were no significant spikes or severe jitters, and the waveform aligned with the action response. After the action was completed, the movement was sustained for another five seconds. The output results indicated that the interaction stability also met our expectations. However, we noted a minor jump in the red waveform during the “V-sign” gesture, which was primarily caused by the instability of the resistance value of the flexible sensor when maintained in a bent state. In the filtering algorithm for the finger-bending degree, the range of this jump did not reach our set threshold, so it was not processed. However, this result did not affect the interaction, and it returned to the normal range in a short time after the jump occurred.

Therefore, from the static interaction experiment, it was found that the motion of the virtual hand essentially aligned with the human hand’s movement, the jitter phenomenon was not pronounced, and, overall, it satisfied the requirements for the system.

Based on the interaction results of static gestures and the calibration of the attitude sensor outputs, dynamic grasping experiments were conducted. In the dynamic tests, we used a geometric sphere as our grasping object. The dynamic grasping tests were divided into three movements: object locking, cradling, and side holding. The interaction results of dynamic grasping are shown in Figure 35.

The dynamic data are shown in Figure 36, where a1–a5 represent the motion data of the five fingers, and the yaw, pitch, and roll represent the motion data of the palm attitude. It can be seen that during the grasping process, the interaction data were relatively stable and smooth.

### 6.3. Debugging of the Overall System

The technologies adopted for communication among the various devices in this system included serial communication, Bluetooth, and Wi-Fi. Firstly, the data glove and forearm communication module transmitted the collected finger-bending and forearm posture motion data to our back-end data management software for data processing through Bluetooth and Wi-Fi modules, respectively. The data management software sent the hand movement information to the execution program for virtual interaction via a virtual serial port, which mapped to our virtual hand. The control board of the robotic hand was connected to the host computer via a serial bus to allow the motion data to drive the robotic hand to complete the corresponding hand movements. The system’s data communication method is illustrated in Figure 37.

The HDT data glove and the forearm module are illustrated in Figure 38.

The five-finger biomimetic robotic hand after the assembly and wiring of all of its components and after being powered up is depicted in Figure 39. After testing, the robotic hand proved to be structurally stable; the servomotor ran smoothly, and the functional requirements of the system were met.

With the master–slave control system and virtual interaction system, the system adjustments were completed. The results of the master–slave control of the hand based on virtual reality are illustrated in Figure 40.

## 7. Conclusions

This study implemented a master–slave control system using hands based on virtual reality technology. By integrating the virtual reality technology with master–slave control technology, it was possible to capture the position, direction, and finger joint angles of a user’s hand in real time. The system allowed for the coordination of a human hand, a mechanical hand, and a virtual hand, and it was tested for its stability and interactivity. The experimental results demonstrated that the system performed excellently in dynamic tracking and can provide users with a natural and agile virtual interaction experience.

This study accomplished the following tasks.

(1)Due to the urgent need for hand protection and the common problems with existing hand master–slave control technologies, we combined virtual reality technology to propose a design scheme for a master–slave control system using hands based on virtual reality technology. We analyzed and identified the four important components of the system, namely, the hand data collection platform, the back-end data management platform, the Unity3D virtual–physical interaction platform, and the master–slave control platform. The working principles and design schemes of each main part were explained in detail.(2)In line with the overall design scheme and requirements, a detailed design and explanation of the system’s hardware structure and constituent components were provided. The hardware included a data glove and a five-fingered bionic mechanical hand, while the software part involved data management software and a virtual interaction program, which eventually resulted in the realization of the debugging and operation of the overall system.(3)In terms of hand posture calculation, we designed a data analysis model for the finger-bending degree and palm posture angles. In terms of data processing, we proposed the integration of complementary filtering based on Kalman filtering, thus fully exploiting the advantages of the two algorithms and compensating for their shortcomings.(4)In research on virtual–real interactions of the hand, we proposed a proxy hand solution, which solved the problem of mutual penetration between the virtual hand and virtual objects during the virtual interaction process. For the judgement of stable grasping, two judgement conditions were proposed, which addressed the non-immersive experience brought by the lack of physical constraints in the virtual world.

Although the system achieved the expected design goals, virtual reality technology involves complex interaction processes, and there are many factors affecting the interaction experience. This study mainly judged the stability of an interaction process based on the forces, speed, and depth of collision between rigid bodies. However, an analysis of other physical constraints between non-rigid bodies was not performed. Therefore, in future research, it will be necessary to continuously expand and conduct in-depth research on this system.

## Figures and Tables

**Figure 1 sensors-23-07107-f001:**
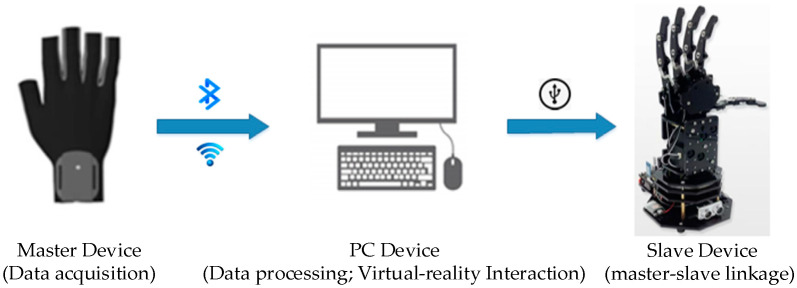
Interactive process of the master–slave virtual–real hand interaction system.

**Figure 2 sensors-23-07107-f002:**
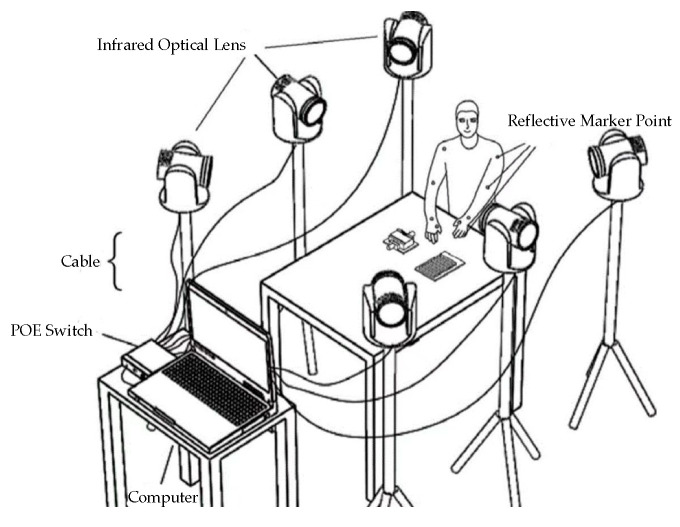
Optical data acquisition system.

**Figure 3 sensors-23-07107-f003:**
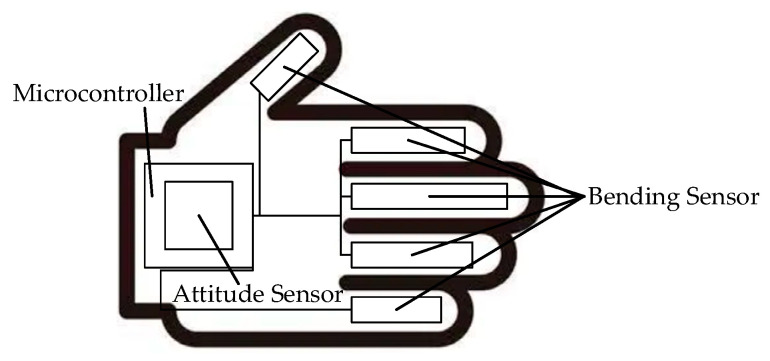
Data glove.

**Figure 4 sensors-23-07107-f004:**
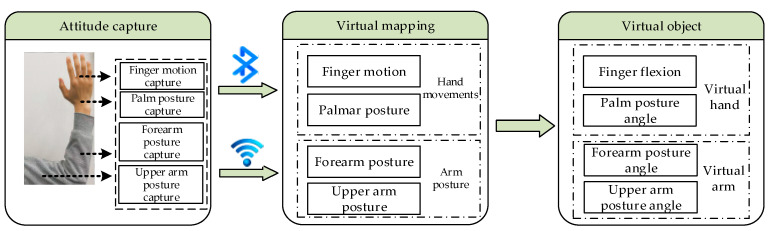
Structure of the system.

**Figure 5 sensors-23-07107-f005:**
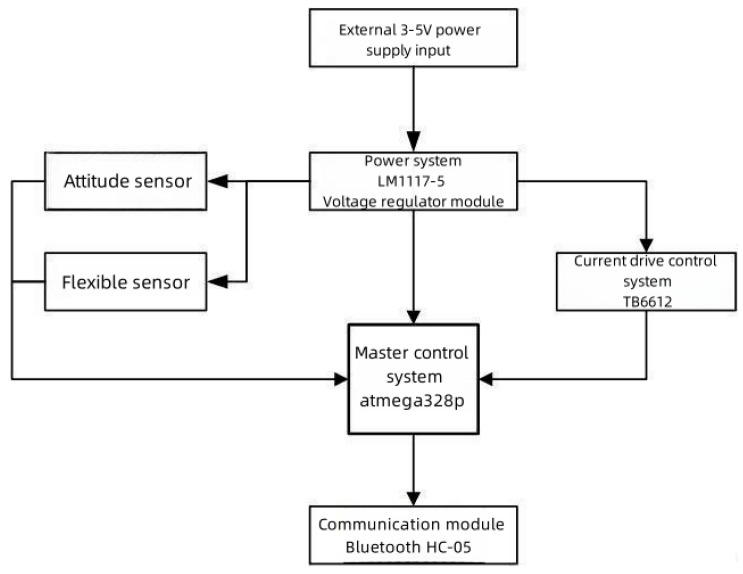
Diagram of the hardware system’s framework.

**Figure 6 sensors-23-07107-f006:**
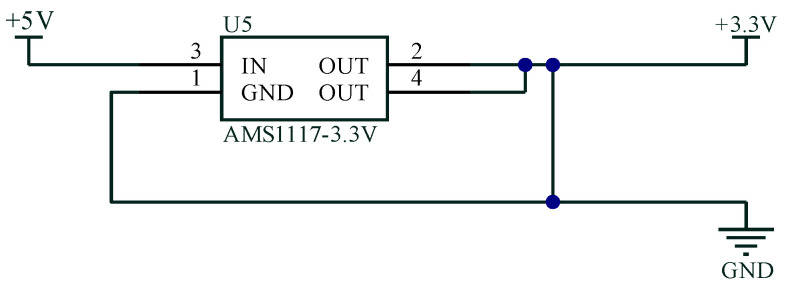
Schematic diagram of the voltage regulation circuit.

**Figure 7 sensors-23-07107-f007:**
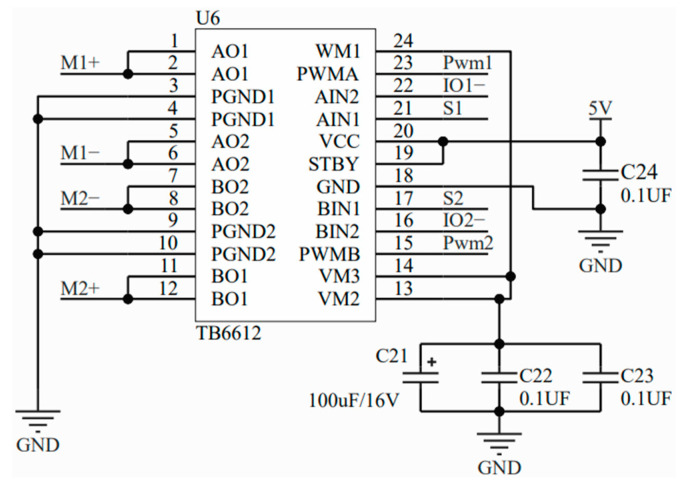
Schematic diagram of the current drive system.

**Figure 8 sensors-23-07107-f008:**
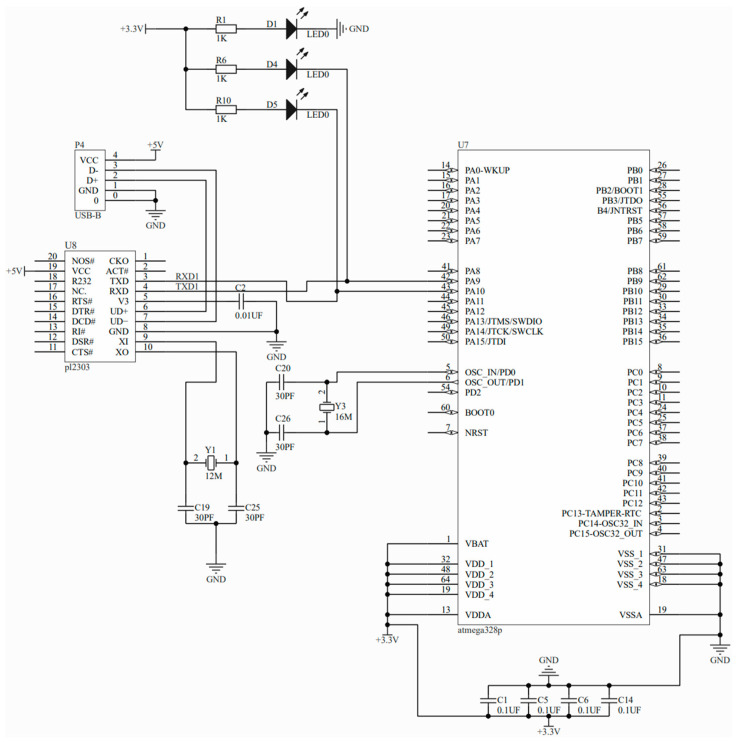
Schematic diagram of the master module.

**Figure 9 sensors-23-07107-f009:**
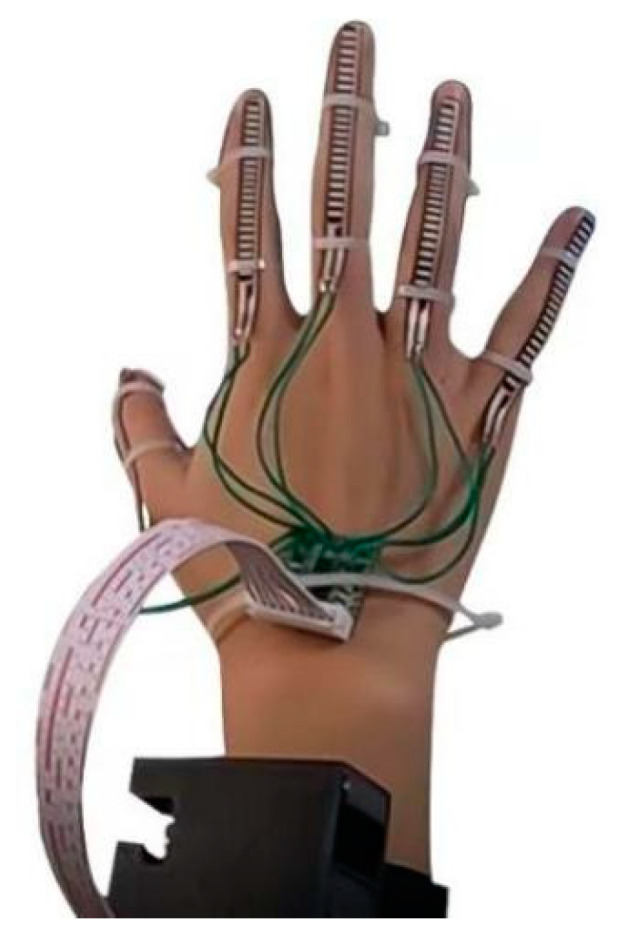
Physical diagram of HDT1.0.

**Figure 10 sensors-23-07107-f010:**
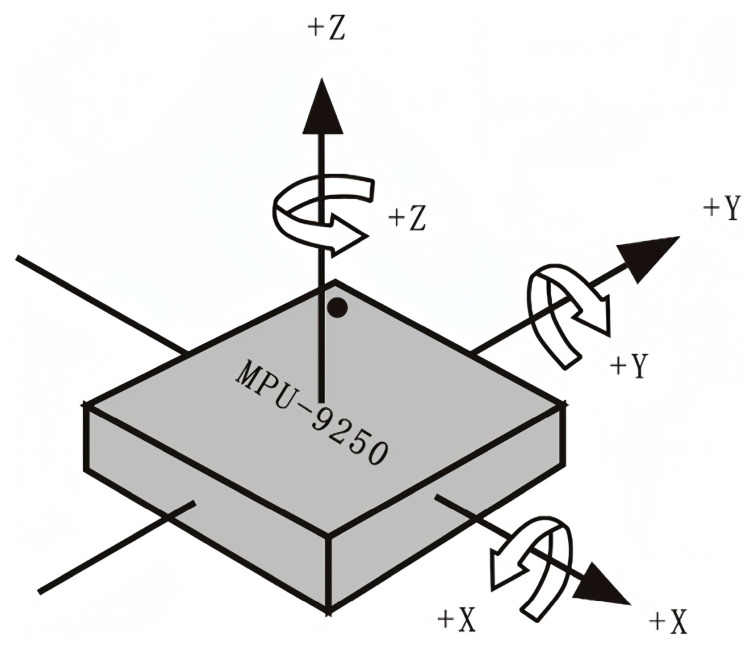
Diagram of the north–east–sky coordinate system.

**Figure 11 sensors-23-07107-f011:**
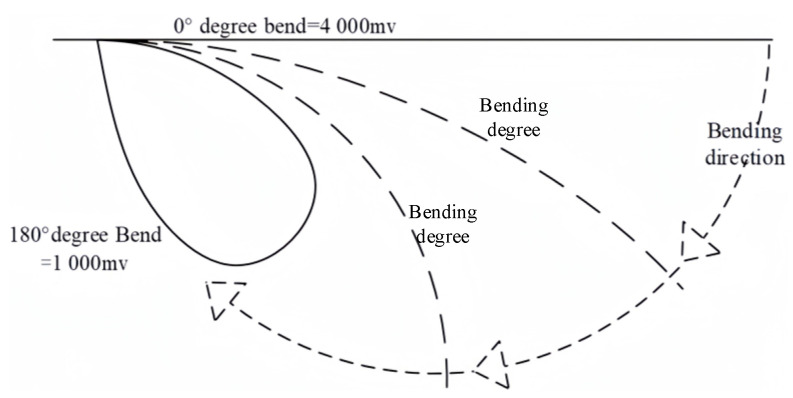
Correlation between output voltage and finger flexion.

**Figure 12 sensors-23-07107-f012:**
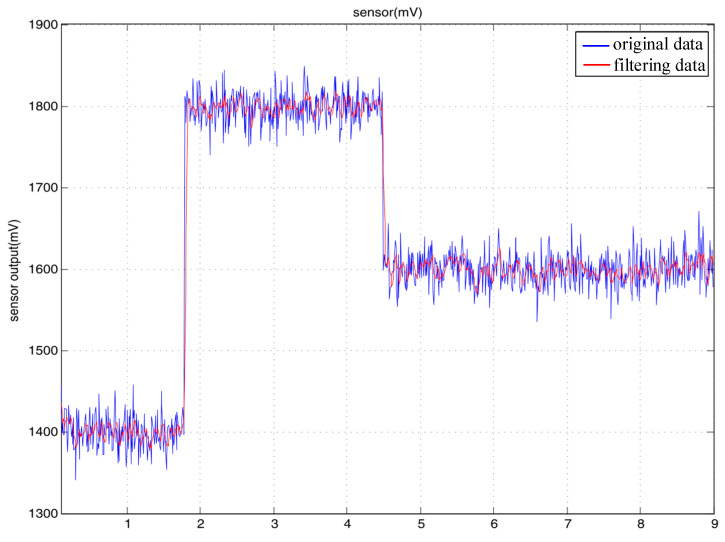
Comparison of the outputs of the flexible sensor.

**Figure 13 sensors-23-07107-f013:**
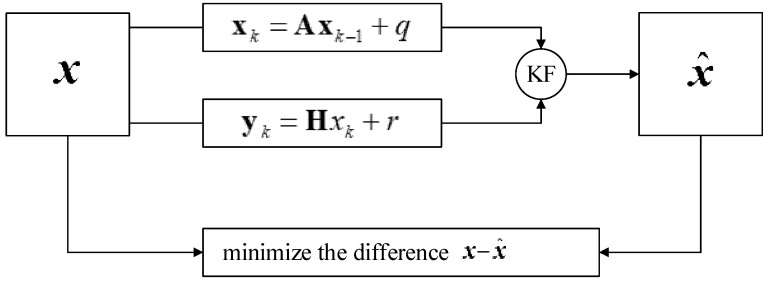
Flowchart of Kalman filtering algorithm.

**Figure 14 sensors-23-07107-f014:**
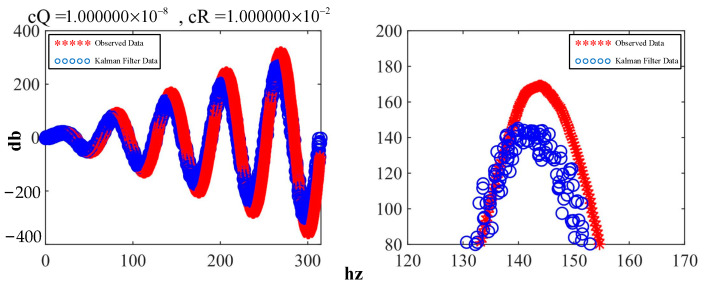
Comparison of the initial effects of the Kalman filter.

**Figure 15 sensors-23-07107-f015:**
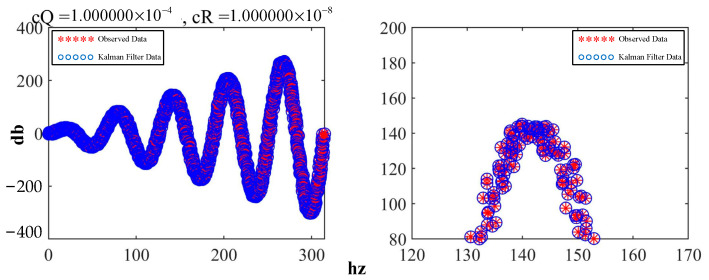
Comparison of the effects after adjusting the parameters.

**Figure 16 sensors-23-07107-f016:**
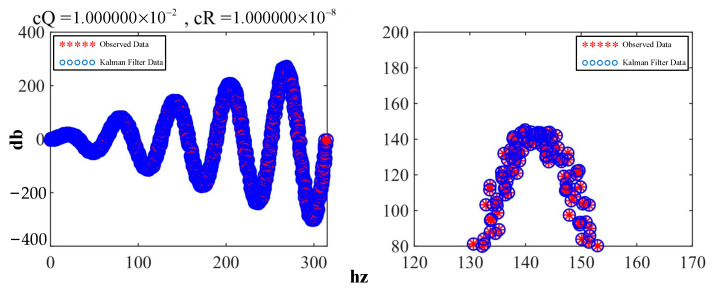
Effect of the Kalman filter with high confidence.

**Figure 17 sensors-23-07107-f017:**
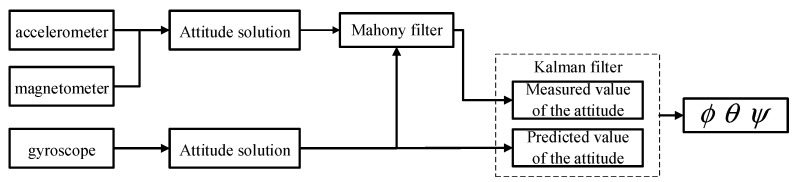
Schematic diagram of the fusion algorithm.

**Figure 18 sensors-23-07107-f018:**
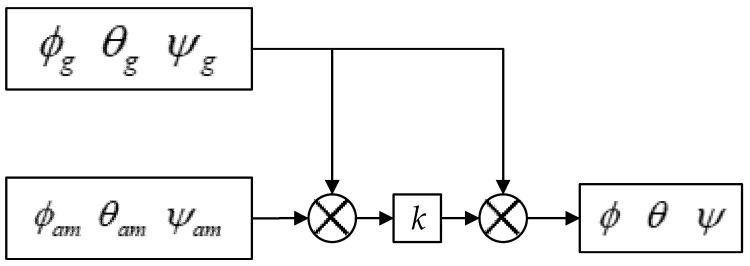
Schematic diagram of the complementary filter algorithm.

**Figure 19 sensors-23-07107-f019:**
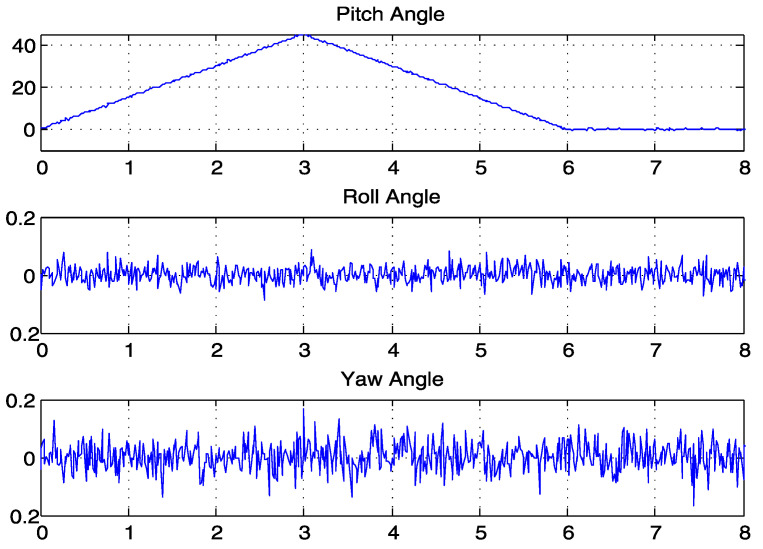
Output waveform of the posture sensor.

**Figure 20 sensors-23-07107-f020:**
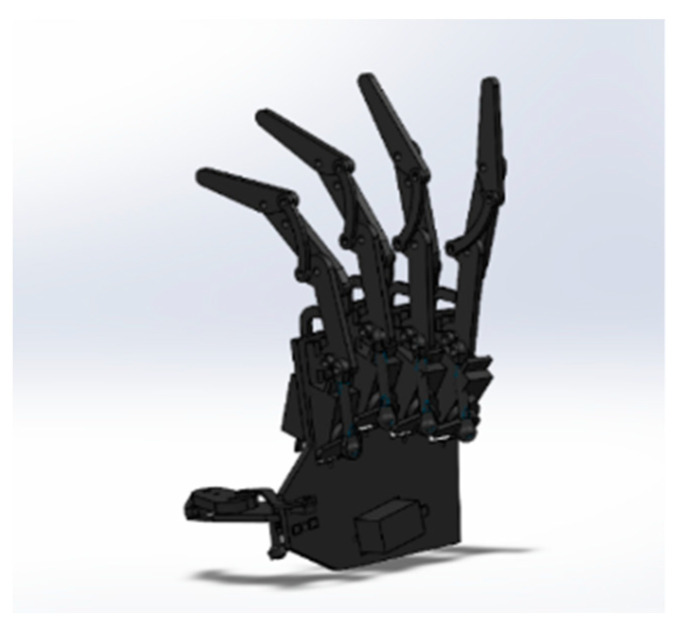
Configuration of the mechanical hand.

**Figure 21 sensors-23-07107-f021:**
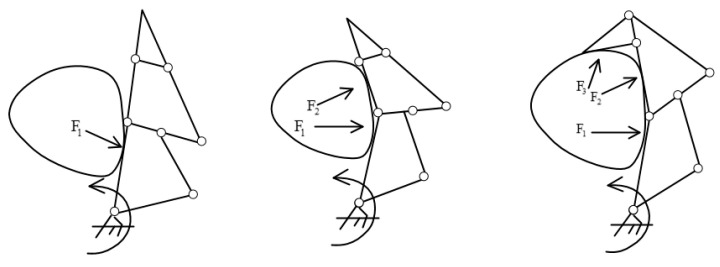
Adaptive grabbing process of the index finger.

**Figure 22 sensors-23-07107-f022:**
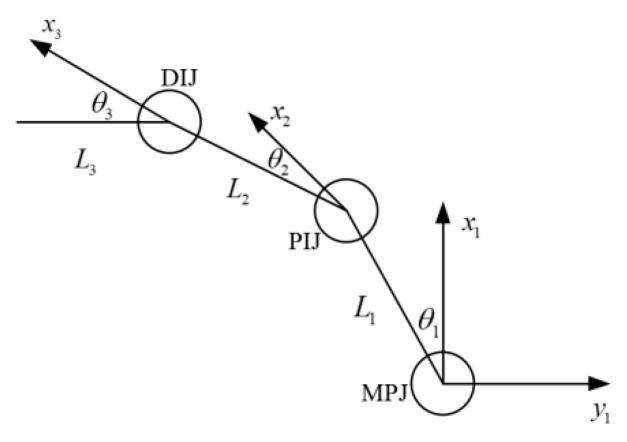
Motion model of the three-link structure.

**Figure 23 sensors-23-07107-f023:**
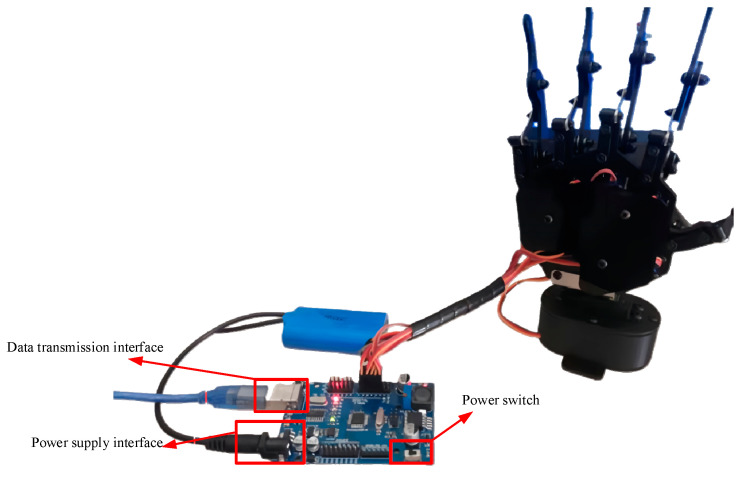
Five-fingered bionic manipulator.

**Figure 24 sensors-23-07107-f024:**
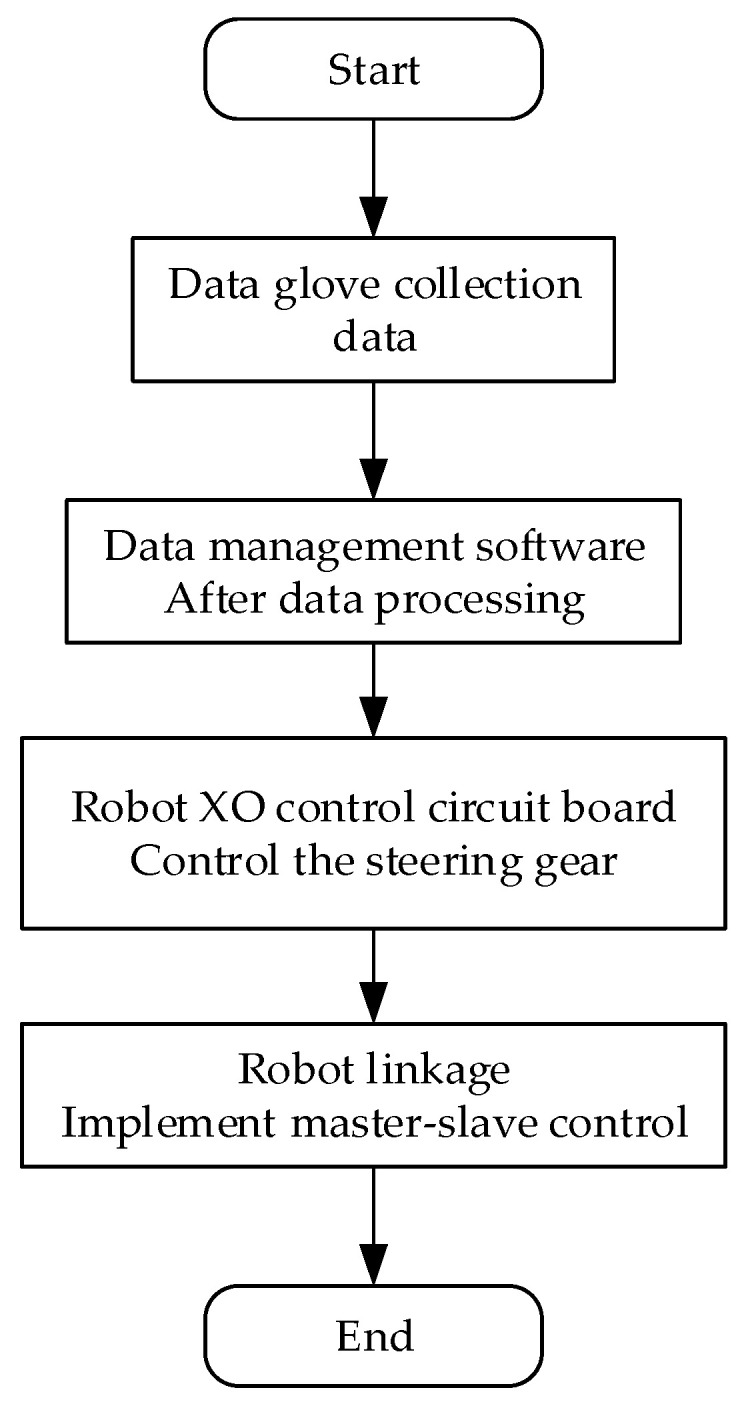
Master–slave control process.

**Figure 25 sensors-23-07107-f025:**
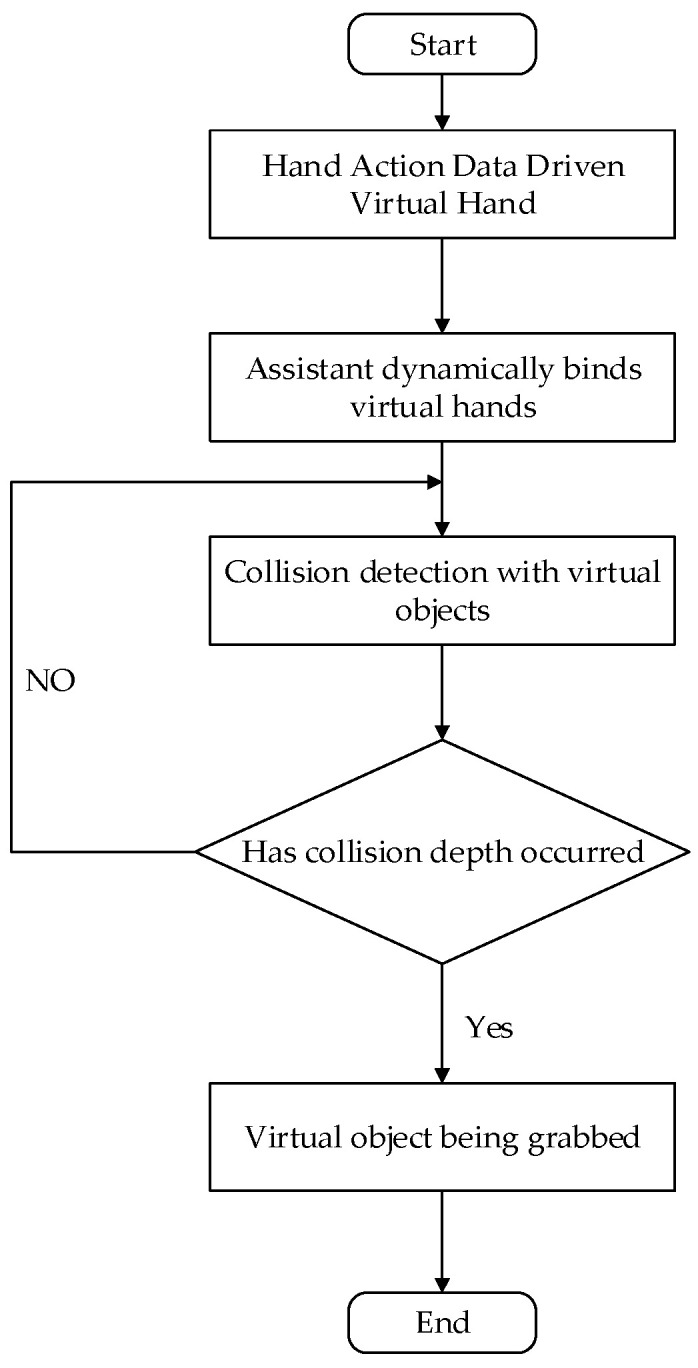
Collision and grabbing process.

**Figure 26 sensors-23-07107-f026:**
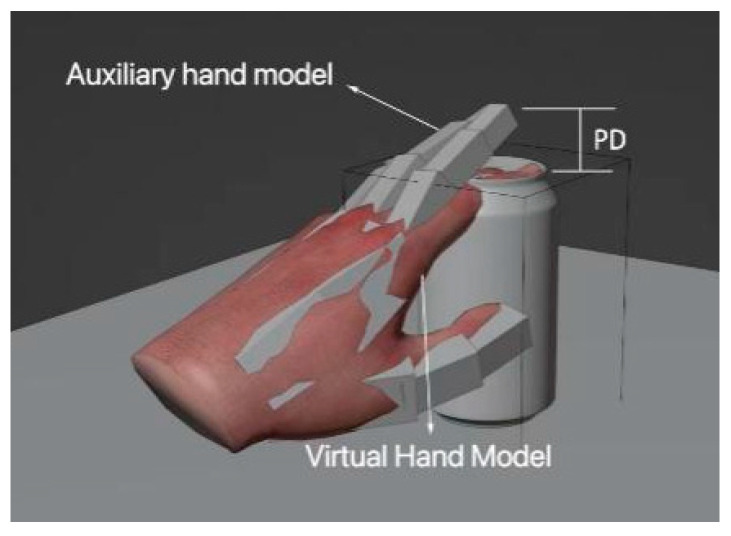
Collision and grabbing principle.

**Figure 27 sensors-23-07107-f027:**
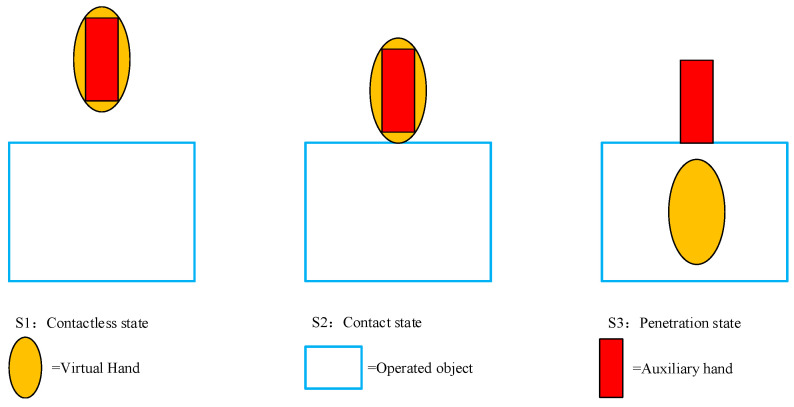
Three states of contact and grabbing.

**Figure 28 sensors-23-07107-f028:**
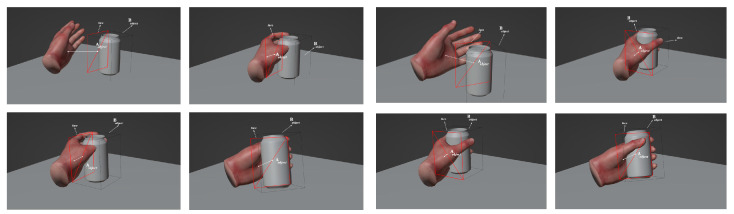
The difference between the collision and grabbing processes of the virtual hand and proxy hand.

**Figure 29 sensors-23-07107-f029:**
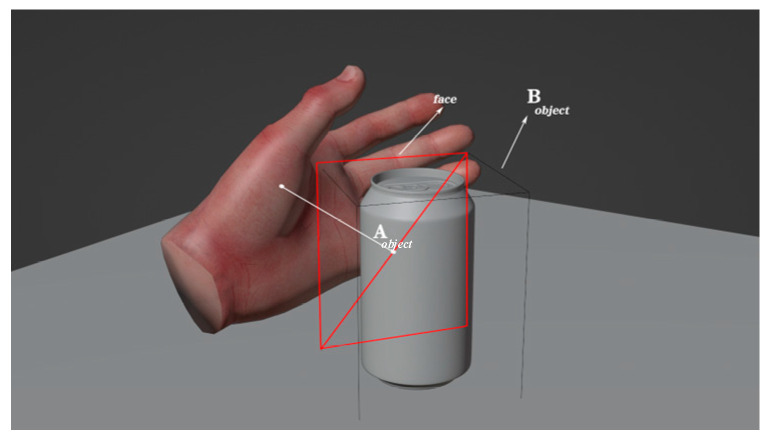
Diagram of grab judgement for the proxy hand.

**Figure 30 sensors-23-07107-f030:**
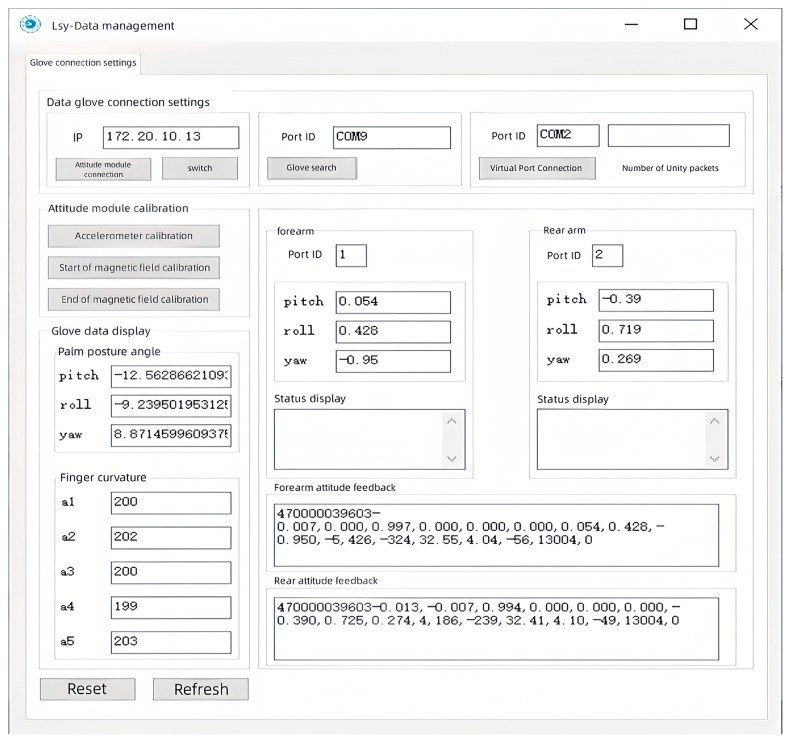
Back-end data management software.

**Figure 31 sensors-23-07107-f031:**
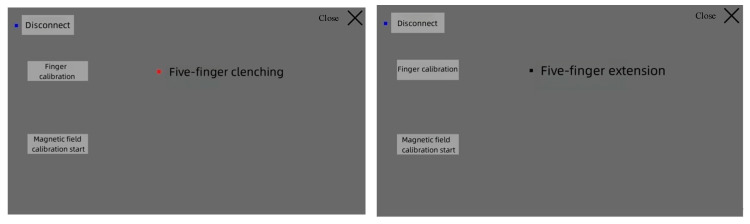
Initial finger calibration.

**Figure 32 sensors-23-07107-f032:**
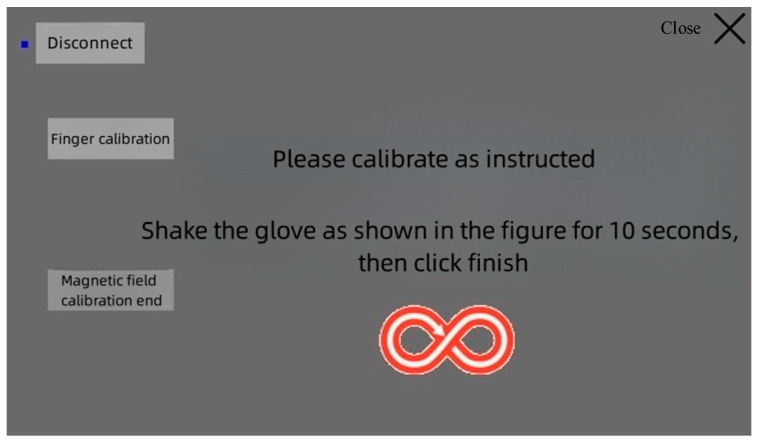
Initial attitude calibration.

**Figure 33 sensors-23-07107-f033:**
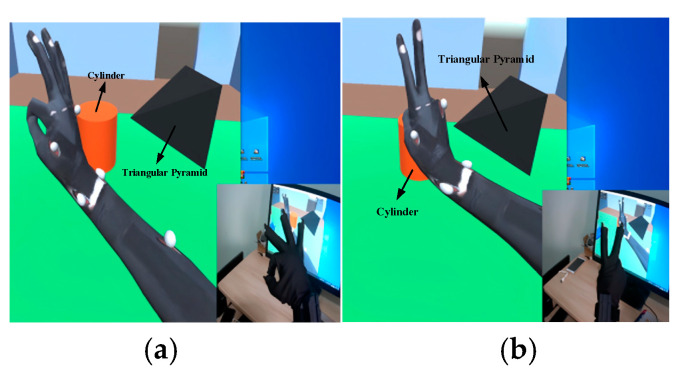
Static gesture interaction. (**a**) Gesture “ok”. (**b**) Gesture “v”.

**Figure 34 sensors-23-07107-f034:**
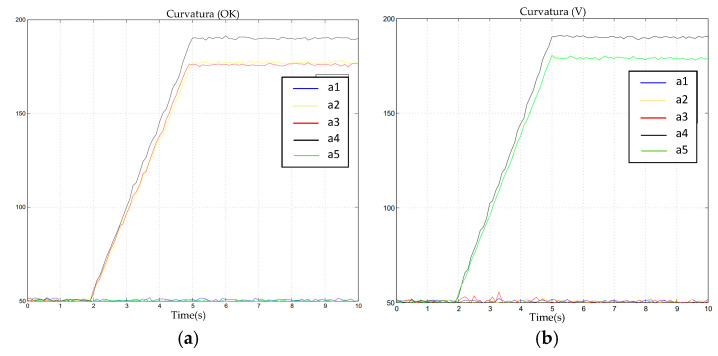
Output waveforms of static gestures. (**a**) Data waveform of gesture “ok”. (**b**) Data waveform of gesture “v”.

**Figure 35 sensors-23-07107-f035:**
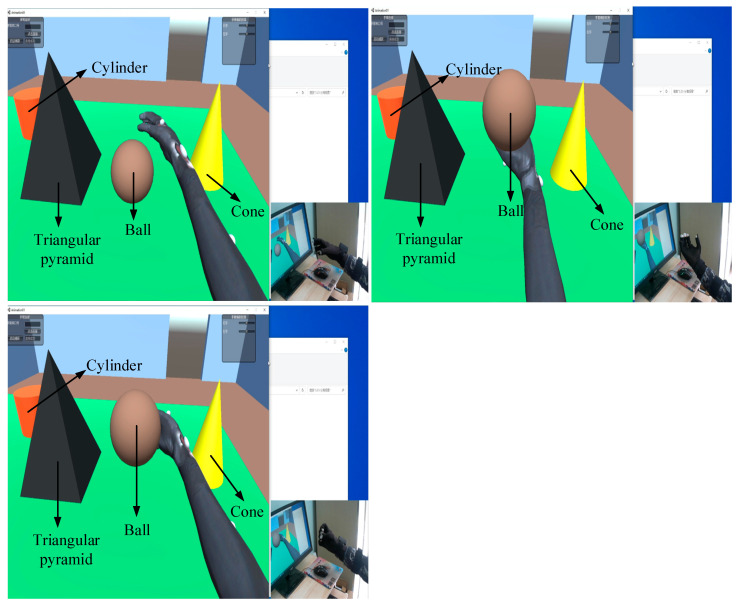
Dynamic grasping interactions.

**Figure 36 sensors-23-07107-f036:**
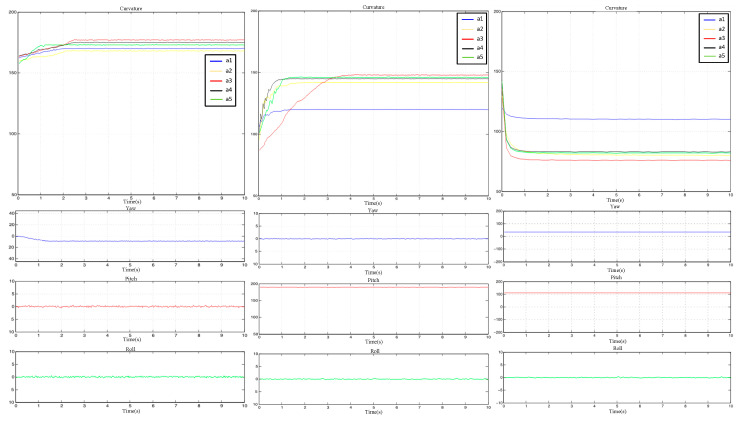
Waveforms of dynamic grasping.

**Figure 37 sensors-23-07107-f037:**
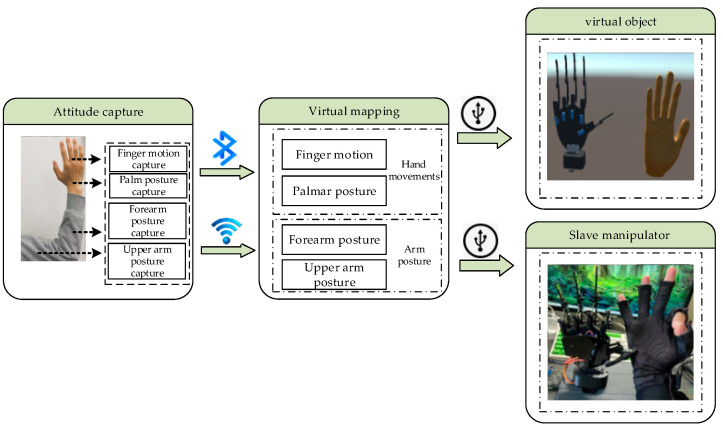
The flow of system data transfer.

**Figure 38 sensors-23-07107-f038:**
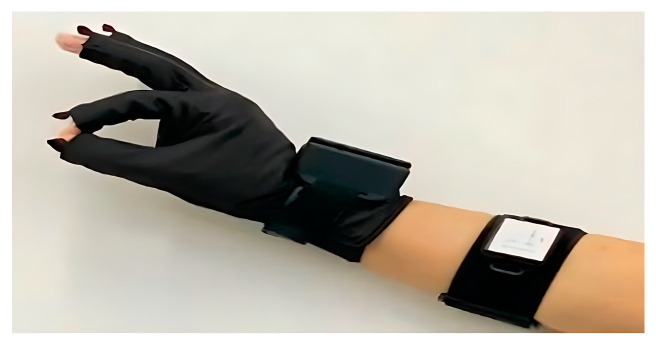
Physical image of the data glove.

**Figure 39 sensors-23-07107-f039:**
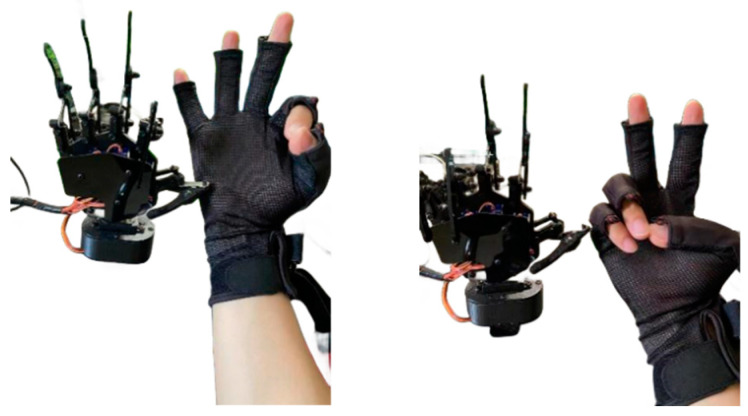
Five-fingered bionic manipulator.

**Figure 40 sensors-23-07107-f040:**
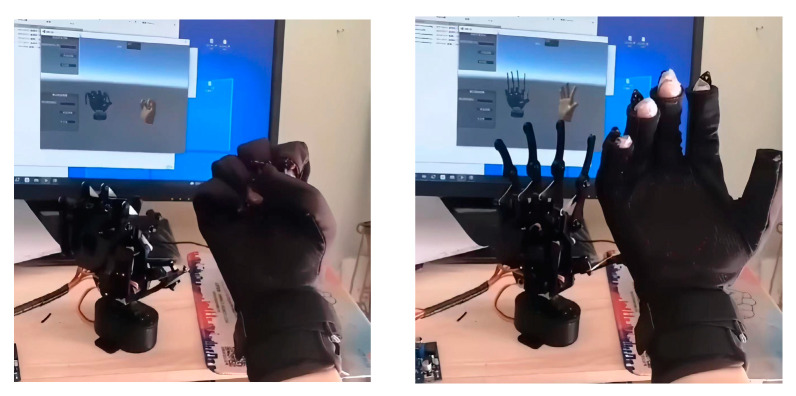
Virtual reality master–slave interactions using a hand.

**Table 1 sensors-23-07107-t001:** Advantages and disadvantages of different manipulators.

Number of Fingers	Number of Joints	Degrees of Freedom of the Fingers	Advantages	Disadvantages
2	3	6	Simple control; no redundancy	Poor grasping effect
3	3	9	Good grasping effect	Poor adaptive grasping
5	3	5	Strong adaptive effect; no redundancy	Average grasping effect
5	3	15	Strong grasping ability; good grasping effect	Complex control, existence of redundancy

**Table 2 sensors-23-07107-t002:** Advantages and disadvantages of different drive systems.

Drive System	Advantages	Disadvantages
Electric drive system	Quick response; high precision of movement	Complex circuit; susceptible to interference
Pneumatic drive system	Simple structure; ample power source	Unsteady movement; has an impact
Hydraulic drive system	Smooth transmission; strong interference resistance	High design and maintenance cost

**Table 3 sensors-23-07107-t003:** Output attitude data.

	Roll1	Pitch1	Yaw1	Roll2	Pitch2	Yaw2
Rotate 45° around the x-axis	45.106	−0.242	0.142	45.096	0.283	0.172
Rotate 45° around the y-axis	−0.176	44.861	0.217	0.109	45.163	0.071
Rotate 45° around the z-axis	0.093	−0.079	45.213	0.366	−0.062	45.122

**Table 4 sensors-23-07107-t004:** D–H parameters.

i	$\theta_{i}$	$d_{i}$	$a_{i}$	$\alpha_{i}$
1	$\theta_{1}$	$-L_{1}\sin\theta_{1}$	$L_{1}\cos\theta_{1}$	0
2	$\theta_{2}$	$-L_{2}\sin\theta_{2}$	$L_{2}\cos\theta_{2}$	0
3	$\theta_{3}$	$-L_{3}\sin\theta_{3}$	$L_{3}\cos\theta_{3}$	0

**Table 5 sensors-23-07107-t005:** Degree of finger bending at 10 s.

	a_1_	a_2_	a_3_	a_4_	a_5_
V	191	1	0	2	185
OK	182	193	186	3	5

## Data Availability

Not applicable.

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
