# Peer review of "Master–Slave Control System for Virtual–Physical Interactions Using Hands"

_sensors, 2023, doi:10.3390/s23167107_

Round 1

Reviewer 1 Report

Figure 17 - low quality, no possibility to analyze the schematization.

Figure 19 - submit in English.

Line 463 - there is a hieroglyph, the meaning is not clear.

I am grateful to the authors. I got incredible pleasure from the article.

Author Response

Dear Reviewer,

I hope this email finds you well. I wanted to express my sincere gratitude for taking the time amidst your busy schedule to review my article. Your insightful feedback and valuable suggestions have been immensely helpful to me. I carefully considered each of your comments and provided responses to them, which can be found in the "Respond reviewer - MDPI" document.

I am pleased to inform you that the revised version of the manuscript has undergone professional proofreading and formatting by the official agency. The updated manuscript is now available for your perusal in the "sensors-2534203-edited-69298" file. I eagerly await your feedback on the revised work.

Once again, thank you for your thoughtful evaluation and support in refining my article. Your contributions have undoubtedly enhanced the quality of the research.

Looking forward to hearing from you soon.

Best regards,
Siyuan Liu

Reviewer 2 Report

This paper is well-organized and written. A few modifications are required before publication.

1.         The abstract can be improved by clarifying the motivation of this study.

2.         The authors are suggested to discuss this paper's structure in the Introduction section.

3.         Related work section is required, and the authors are suggested to analyze and compare the proposed approach with these existing studies.

4.         The implementation environment should be added in Section 6.

5.         The description of Figure 21 is insufficient and the authors are suggested to clarify the output results.

6.         The conclusion section can be improved by discussing the limitation of this approach and some future research directions.

7.         The English editing can be improved.

Extensive editing of English language required

Author Response

(The authors gave the same response as above.)

Reviewer 3 Report

The comments can be found in the attached file. 

The comments can be found in the attached file. 

Author Response

(The authors gave the same response as above.)

Reviewer 4 Report

I have carefully read your paper and evaluated it. The paper proposes an innovative master-slave control system for hand virtual real interaction, designs an HDT data glove and forearm pose detection module, and uses Kalman pose fusion algorithm and collision grasping algorithm to achieve synchronous motion of virtual hands and robotic hands and stable grasping of virtual objects. The paper shows your professional knowledge and technical capabilities in this field, making a certain contribution to promoting the development.

However, in my opinion, the paper still has some shortcomings that need to be improved before it is officially published. Here are some comments as follow:

1. The paper did not provide sufficient details regarding the HDT data glove and forearm posture detection module, such as sensor specifications, calibration methods, and error analysis. And explain how you ensure the accuracy and reliability of the data collection process, as well as how to handle possible noise or interference.

2. The paper does not explain how you choose or adjust the parameters of the Kalman filter algorithm, and what are the advantages, disadvantages or limitations of using this algorithm for attitude estimation. Please elaborate on these issues in the article and provide some theoretical analysis or experimental results to support your choice. In addition, please compare or discuss with other possible methods, such as complementary filtering, extended Kalman filter filtering or unscented Kalman filter filtering.

3. The paper does not provide any quantitative or qualitative results to demonstrate the effectiveness and efficiency of your proposed system in hand virtual real interaction. You can add some experimental or evaluation sections to the paper, using appropriate indicators or standards to measure or compare the differences or advantages of your system in accuracy and stability compared to other existing systems or benchmark methods. Meanwhile, please consider different scenarios, such as different hand sizes, shapes, gestures, or objects, and analyze whether your system can work properly or adapt to changes in these scenarios.

4. There are also Chinese appearing in the paper, such as line 463. It is recommended to modify Figure 19 to English.

In summary, I believe that your paper has some innovation and value, but it also needs further improvement. I suggest that major revisions to this paper based on my comments.

none

Author Response

(The authors gave the same response as above.)

Reviewer 5 Report

In this paper, a virtual-physical hand interaction master-slave control system is designed, which can capture the position, direction and finger joint Angle of the user's hand in real time, and realize synchronous interaction between the virtual hand and the manipulator through limiting, jitter reduction and complementary filtering combined with Kalman filtering algorithm.Satisfy the requirement of the system under different working condition of precision.This system has potential application in virtual reality field.

The following questions remain:

1.Line 157:The format is wrong, the formula is misaligned.

2.Part 4.1(Line 192)and Part 4.2(Line 208):Are there any metrics for estimation accuracy, or providing comparisons with other algorithms?

3.Line 304:Whether the results of the simulation model can be supplemented.

4.Part 7(Line 520):Since it is interactive,Whether timeliness related tests are required.

This paper designed a virtual hand - physical interaction master-slave control system, by improving the algorithm, implements the drive virtual hand and synchronous interaction of the manipulator.But there are some small problems.Therefore, the suggestion of minor revision is given.

Minor editing of English language required

Author Response

(The authors gave the same response as above.)

Round 2

Reviewer 2 Report

The authors have incorporated all my concerns. There are no further requirements from my side.

Reviewer 4 Report

The author made revisions to the paper according to my suggestions and I suggested to publish it.

none